# *Mycobacterium tuberculosis* PhoP integrates stress response to intracellular survival by regulating cAMP level

Hina Khan[1‡], Partha Paul[1†], Harsh Goar[1†§], Bhanwar Bamniya[1,2†], Navin Baid[1#], Dibyendu Sarkar[1,2*]

[1]CSIR, Institute of Microbial Technology, Chandigarh, India; [2]Academy of Scientific and Innovative Research, Ghaziabad, India

*For correspondence:
dibyendu@imtech.res.in

[†]These authors contributed equally to this work

Present address: [‡]Department of Physiology, Michigan State University, Interdisciplinary Science and Technology Building, Lansing, United States; [§]Department of Medicine, Division of Hematology-Oncology UT Southwestern Medical Center, Dallas, United States; [#]Department of Biosciences and Bioengineering Indian Institute of Technology Bombay, Mumbai, India

Competing interest: The authors declare that no competing interests exist.

**eLife assessment**

This **important** study describes how PhoP regulates cyclic-AMP production in the human pathogen *Mycobacterium tuberculosis*. The authors provide **convincing** evidence that PhoP acts as a repressor of the cyclic-AMP-specific phosphodiesterase, Rv0805, which can degrade cyclic-AMP. The article has addressed all outstanding comments, and the work will be of interest to bacteriologists.

**Abstract** Survival of *Mycobacterium tuberculosis* within the host macrophages requires the bacterial virulence regulator PhoP, but the underlying reason remains unknown. 3′,5′-Cyclic adenosine monophosphate (cAMP) is one of the most widely used second messengers, which impacts a wide range of cellular responses in microbial pathogens including *M. tuberculosis*. Herein, we hypothesized that intra-bacterial cAMP level could be controlled by PhoP since this major regulator plays a key role in bacterial responses against numerous stress conditions. A transcriptomic analysis reveals that PhoP functions as a repressor of cAMP-specific phosphodiesterase (PDE) Rv0805, which hydrolyzes cAMP. In keeping with these results, we find specific recruitment of the regulator within the promoter region of *rv0805* PDE, and absence of *phoP* or ectopic expression of *rv0805* independently accounts for elevated PDE synthesis, leading to the depletion of intra-bacterial cAMP level. Thus, genetic manipulation to inactivate PhoP-*rv0805*-cAMP pathway decreases cAMP level, stress tolerance, and intracellular survival of the bacillus.

## Introduction

*Mycobacterium tuberculosis*, the causative agent of pulmonary tuberculosis, encounters diverse environmental conditions during infection, persistence, and transmission of the disease (*Ehrt et al., 2018*; *Ernst, 2012*; *Russell, 2011*). However, the pathogen is exceptionally capable of adjusting and surviving within diverse host environments. Mycobacterial adaptive response to different stages of infection is achieved via fine-tuning of regulation of gene expression using an extensive repertoire of more than 100 transcriptional regulators, 11 two-component systems, 6-serine-threonine protein kinases, and 13 alternative sigma factors, suggesting that a very complex transcriptional program is important for *M. tuberculosis* pathogenesis. While the regulation of gene expression as a consequence of interaction of tubercle bacilli with its immediate environment remains critically important (*Rohde et al., 2007*), defining these regulatory pathways represents a major challenge in the field.

3′,5′-cyclic adenosine monophosphate (cAMP), one of the most widely used second messengers, impacts a wide range of cellular responses in microbial pathogens, including *M. tuberculosis* (*McDonough and Rodriguez, 2011*). The bacterial genome encodes at least 15 adenylate cyclases

(ACs), including one of the ACs (Rv0386) that is required for virulence (*Agarwal et al., 2009*), and multiple cAMP-regulated effector proteins (*Banerjee et al., 2015*; *Johnson and McDonough, 2018*; *McDonough and Rodriguez, 2011*; *Shenoy and Visweswariah, 2006*). cAMP levels are elevated upon infection of macrophages by pathogenic mycobacterium (*Bai et al., 2009*), and addition of exogenous cAMP was shown to influence mycobacterial protein expression (*Gazdik and McDonough, 2005*). While intra-bacterial cAMP responds to macrophage environment and carries out specific functions like gene expression and protein function under host-related conditions (*Bai et al., 2011*; *Bai et al., 2009*; *Johnson et al., 2017*; *Johnson and McDonough, 2018*; *Knapp and McDonough, 2014*), secreted cAMP controls macrophage signaling (*Agarwal et al., 2009*; *Agarwal et al., 2006*; *Gazdik et al., 2009*; *Johnson et al., 2017*; *Nambi et al., 2013*; *Ranganathan et al., 2016*; *Rittershaus et al., 2018*). cAMP signaling remains essential to *M. tuberculosis* pathogenesis. Agarwal and colleagues had shown that a burst in synthesis of cAMP upon infection of macrophages improved bacterial survival by interfering with host signaling pathways (*Agarwal et al., 2009*). In keeping with this, anti-tubercular compounds that interfere with mycobacterial cAMP levels also impact intracellular growth of mycobacteria (*Bai et al., 2011*; *Johnson et al., 2017*; *Knapp et al., 2015*; *Nambi et al., 2013*; *Rittershaus et al., 2018*; *VanderVen et al., 2015*; *Wilburn et al., 2022*). Together, these results underscore the significance of cAMP signalling in mycobacteria.

cAMP signaling is controlled at the transcriptional level. Previous in silico studies had identified 2 out of 10 predicted nucleotide-binding proteins of *M. tuberculosis* as members of the CRP/FNR superfamily of transcriptional regulators (*McCue et al., 2000*). These are CRP (cyclic AMP receptor protein), encoded by *rv3676* and CMR (cAMP macrophage regulator), encoded by *rv1675c*, respectively. Of these two, CRP becomes activated upon cAMP binding and functions as a global regulator of ~100 genes (*Agarwal et al., 2006*; *Bai et al., 2005*; *Rickman et al., 2005*). Consequently, deletion of the *crp* locus significantly impairs mycobacterial growth and attenuates virulence of the bacilli in a mouse model (*Rickman et al., 2005*). In contrast, CMR is necessary for regulated expression of genes involved in virulence and persistence, including members of the dormancy regulon (*Gazdik et al., 2009*; *Ranganathan et al., 2016*; *Smith et al., 2017*). These results strongly suggest that cAMP homeostasis, that is, a balance between cAMP synthesis by ACs and cAMP degradation by phosphodiesterases (PDEs), contributes to rapid adaptive response of mycobacteria in a hostile intracellular environment (*Johnson and McDonough, 2018*; *McDonough and Rodriguez, 2011*). However, very little is known about the underlying mechanisms of regulation of mycobacterial cAMP level.

*M. tuberculosis* encounters a hostile environment within the host. Among the hostile conditions are the acidic pH stress and exposure to host immune effectors such as NO (*Nathan and Shiloh, 2000*; *Rustad et al., 2009*; *Wayne and Sohaskey, 2001*). A growing body of evidence connects virulence-associated mycobacterial *phoP* locus with varying environmental conditions, including acid stress (*Abramovitch et al., 2011*; *Bansal et al., 2017*; *Tan et al., 2013*), heat-shock (*Sevalkar et al., 2019*; *Singh et al., 2014*), and integration of acid stress response to redox homeostasis (*Baker et al., 2019*; *Baker et al., 2014*; *Goar et al., 2022*). Disruption of *phoP*, the gene encoding the response regulator of the PhoPR two-component signal transduction system (*Gupta et al., 2006*), significantly reduces in vivo multiplication of the bacilli (*Walters et al., 2006*), suggesting that PhoPR remains essential for virulence (*Pérez et al., 2001*). Moreover, a mutant lacking this system shows a significantly lowered synthesis of cell wall components diacyltrehaloses, polyacyltrehaloses, and sulfolipids, specific to pathogenic mycobacterial species (*Gonzalo Asensio et al., 2006*; *Goyal et al., 2011*; *Walters et al., 2006*). In fact, the significant attenuation of *phoPR* deletion strain forms the basis of the mutant being considered in trials as a vaccine strain (*Arbues et al., 2013*). Recent transcriptomic analyses revealed that approximately 2% of the H37Rv genome is regulated by PhoP (*Solans et al., 2014*; *Walters et al., 2006*), and mycobacterial gene expression in response to acidic pH significantly overlaps with the PhoP regulon (*Rohde et al., 2007*). Consistent with this, a large subset of PhoPR-regulated low-pH-inducible genes are induced immediately following *M. tuberculosis* phagocytosis and remain induced during macrophage infection (*Gonzalo Asensio et al., 2006*; *Martin et al., 2006*; *Pérez et al., 2001*; *Walters et al., 2006*). Additional evidence, coupled with more recent results, suggests that during the onset of macrophage infection PhoPR activation is linked to acidic pH and the available carbon source, suggesting a physiological link between pH, carbon source, and macrophage pathogenesis (*Baker et al., 2019*).

In this study, we hypothesized that intra-mycobacterial cAMP level could be determined by PhoP since the major regulator has been implicated in the regulation of bacterial responses against numerous stress conditions, many of which function as signals to activate cAMP synthesizing diverse ACs (*Knapp and McDonough, 2014*). Our results connect virulence regulator PhoP with intra-mycobacterial cAMP level. We discovered that PhoP regulates the expression of cAMP-specific phosphodiesterase *rv0805*, which hydrolyzes cAMP. To further probe the regulation, we demonstrate that under the condition that activates PhoP-PhoR system, PhoP in a PhoR-dependent manner represses the transcription of *rv0805* through direct DNA binding at the upstream regulatory region. These observations account for a consistently lower level of cAMP in a PhoPR-deleted *M. tuberculosis* H37Rv (*phoPR*-KO) relative to the WT bacilli and establishes the molecular mechanism of regulation of cAMP level, absence of which strikingly impacts phagosome maturation, and reduces mycobacterial survival within macrophages and mice. Together, the newly identified mechanism of regulation of cAMP level allows intra-phagosomal survival and growth program of mycobacteria.

## Results

### Intra-mycobacterial cAMP level is regulated by the *phoP* locus

We compared cAMP levels of WT and *phoPR*-KO mutant (lacking both the single copies of *phoP* and *phoR* genes), grown under normal, NO stress, and acid stress conditions (*Figure 1A*). *phoPR*-KO showed a significantly lower level of cAMP relative to the WT bacilli, both under normal and stress conditions. Complementation of the mutant (Compl.) could restore cAMP to the WT level. Under normal conditions and NO stress, a higher cAMP level in the complemented strain under NO stress is possibly attributable to reproducibly higher *phoP* expression in the complemented mutant under specific stress conditions (*Khan et al., 2022*). Because bacterial growth often varies under stress conditions, and growth inhibition can influence cAMP level, we compared the viability of mycobacterial strains under normal and indicated stress conditions (conditions of cAMP measurements) by determining the bacterial CFU (*Figure 1—figure supplement 1*). Note that for in vitro viability under specific stress conditions, the indicated mycobacterial strains were grown to the mid-log phase (OD$_{600}$ 0.4–0.6) and exposed to acidic media (7H9 media, pH 4.5 [*Gouzy et al., 2021*]) for further 2 hr at 37°C. Likewise, for NO stress, cells grown to the mid-log phase were exposed to 0.5 mM DetaNonoate for 40 min. Our results suggest that WT and *phoPR*-KO under carefully controlled stress conditions display comparable viability, indicating that the variation in viable cell counts of the mutant under specific stress conditions does not account for lower cAMP level. From these results, we conclude that PhoP is controlling cAMP level mainly in stressed cells.

To investigate the possibility that *phoPR*-KO secretes out more cAMP and, therefore, shows a lower cytoplasmic level, we compared cAMP secretion of WT and the mutant (*Figure 1B*). The WT and mutant strains were grown as described in 'Materials and methods' and culture filtrates (CF) as well as cell lysates (CL) were collected as described previously (*Anil Kumar et al., 2016*). Our results demonstrate that the mutant reproducibly secretes lower amount of cAMP relative to the WT bacilli, and cAMP secretion is fully restored in the complemented mutant (Compl.). As the fold difference of secretion (~1.25-fold) is not much different relative to the fold difference in intra-mycobacterial cAMP level (~2-fold), we suggest that lower cAMP level of the mutant is not due to its higher efficacy of cAMP secretion. *Figure 1C* confirms the absence of autolysis of mycobacterial cells as GroEL2, a cytoplasmic protein, was undetectable in the CFs.

### PhoP functions as a repressor of *rv0805*

We next investigated the role of the *phoP* locus on the expression of mycobacterial ACs and PDEs, which synthesize and degrade cAMP, respectively (*Figure 2A*). The selection of ACs and PDEs was based on two key points. First, we have chosen ACs, which are activated by known signals (*Knapp and McDonough, 2014*). Second, we reasoned that the previously reported ACs were activated under environmental conditions, which are linked to mycobacterial *phoP* locus (*Bansal et al., 2017*; *Goar et al., 2022*). Our RT-qPCR results using gene-specific primer pairs (*Supplementary file 1a*) suggest that expression of ACs, including *rv0386*, *rv1264*, *rv1647*, and *rv2488c* (*Agarwal et al., 2009*; *Dass et al., 2008*; *Dittrich et al., 2006*; *Knapp and McDonough, 2014*), does not appear to be regulated by the *phoP* locus. However, a significant activation of AC *rv0891c* (4 ± 0.05-fold) and repression of

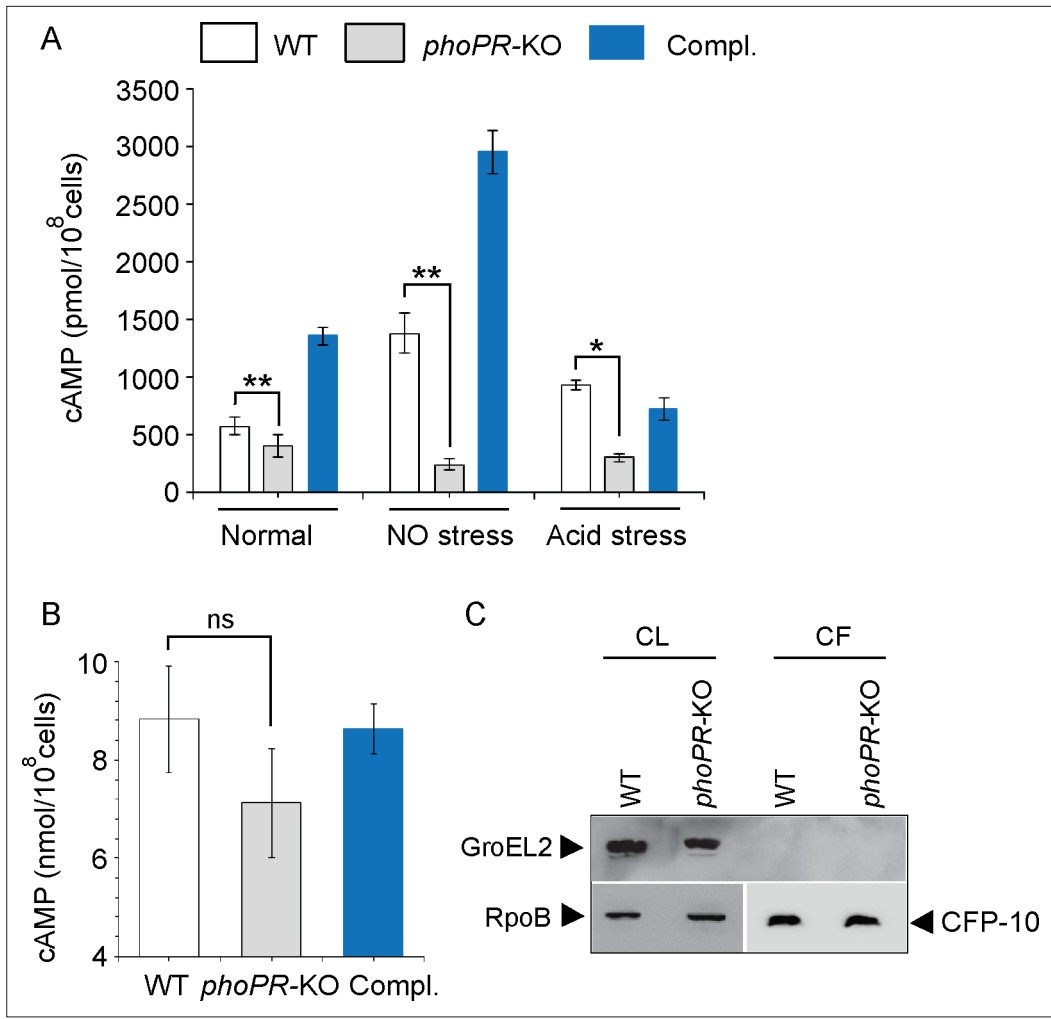

**Figure 1.** PhoP contributes to the maintenance of mycobacterial 3',5-cyclic adenosine monophosphate (cAMP) level. (**A**) Intra-mycobacterial cAMP levels were determined by a fluorescence-based assay as described in 'Materials and methods', and compared for indicated mycobacterial strains, grown under normal or specific stress conditions. For acid stress, mycobacterial strains were initially grown to the mid-log phase ($OD_{600}$ 0.4–0.6), and then transferred to acidic pH (7H9 media, pH 4.5) for further 2 hr of growth at 37°C. For NO stress, cells grown to the mid-log phase were exposed to 0.5 mM DetaNonoate for 40 min. The data represent average values from three biological repeats (*$p \leq 0.05$, **$p \leq 0.01$). (**B**) To compare the secretion of cAMP by WT and *phoPR*-KO, cAMP levels were also determined in the corresponding culture filtrates (CF) (ns., non- significant). (**C**) Immunoblotting analysis of 10 µg of cell lysates (CL) and 20 µg of CF of indicated *M. tuberculosis* strains. α-GroEL2 was used as a control to verify cytolysis of cells, CFP-10 detected as a secreted mycobacterial protein in the CFs, and RpoB used as the loading control.

The online version of this article includes the following source data and figure supplement(s) for figure 1:

**Source data 1.** cAMP estimation in WT, phoPR-KO and the complemented mutant.

**Source data 2.** Quantification of cAMP secretion of WT, phoPR-KO and the complemented mutant.

**Source data 3.** GroEL2 levels in cell lysates of WT-H37Rv and phoPR-KO.

**Source data 4.** GroEL2 levels in cell lysates of WT-H37Rv and phoPR-KO.

**Source data 5.** Anti- RpoB blot as a loading control of comparable amount of cell lysates.

**Source data 6.** Anti-RpoB blot as a loading control of comparable amount of cell lysates.

**Source data 7.** CFP-10 in culture filtrates of H37Rv and phoPR-KO.

**Source data 8.** CFP-10 in the culture filtrates of H37Rv and phoPR-KO.

*Figure 1 continued on next page*

*Figure 1 continued*

**Figure supplement 1.** Defined and indicated stress conditions do not influence in vitro growth of mycobacterial strains.

**Figure supplement 1—source data 1.** CFU of indicated mycobacterial strains under normal and stress conditions.

phosphodiesterase *rv0805* expression (6.5 ± 0.7-fold), respectively, were dependent on the *phoP* locus. Although Rv0891c was suggested as one of the *M. tuberculosis* H37Rv ACs, the protein lacks most of the important residues conserved for the AC family of proteins (*Zaveri et al., 2021*). On the other hand, *rv0805* encodes for a cAMP-specific PDE, present only in slow-growing pathogenic *M. tuberculosis* (*Matange et al., 2013*; *Shenoy et al., 2007*; *Shenoy et al., 2005*). Although the expression of *rv0805* was restored at the level of WT in the complemented mutant (Compl.), we observed poor restoration of the expression of *rv0891c* in the Compl. strain. Further, expression of PDE *rv1339* that contributes to mycobacterial cAMP level (*Thomson et al., 2022*) remains unaffected by the *phoP* locus. Therefore, we focussed our attention on the biological significance of PhoP-dependent regulation of *rv0805*. It should be noted that expressions of *rv1357c* and *rv2837c*, encoding the PDEs for cyclic di-GMP (*Flores-Valdez et al., 2015*) and cyclic di-AMP (*Valadares and Woo, 2017*), respectively, remained unchanged in *phoPR*-KO.

As we reproducibly observed activation of *rv0805* expression in *phoPR*-KO (relative to WT), we investigated whether acidic pH conditions under which *phoPR* system is activated (*Abramovitch et al., 2011*; *Bansal et al., 2017*) impact the expression of *rv0805* in WT bacilli (*Figure 2B*). Our results show that the repression of *rv0805* is significantly higher in WT grown under acidic conditions (pH 4.5) relative to the normal conditions (pH 7.0) of growth. This observation is consistent with RNA-seq data displaying significant downregulation of *rv0805* in WT bacilli grown under acidic pH conditions relative to the normal conditions of growth (GEO accession number: GSE180161). As a control, expression of PDE *rv1339*, which also degrades cAMP, remains unaffected under acidic conditions of growth. The finding that acidic pH (pH 4.5) conditions of growth promoted PhoP-dependent repression of *rv0805* prompted us to investigate whether PhoP directly binds to *rv0805* promoter. To this end, we next examined in vivo recruitment of PhoP within the *rv0805* promoter by chromatin immunoprecipitation (ChIP) experiments (*Figure 2C*). In this assay, formaldehyde-cross-linked DNA-protein complexes of growing *M. tuberculosis* cells were sheared to generate fragments of average size ≈500 bp. Next, ChIP experiments utilized anti-PhoP antibody, and IP DNA was analyzed by real-time qPCR relative to a mock sample (without antibody as a control) using FPrv0805up/RPrv0805up as the primer pair (*Supplementary file 1a*). Our results show that under normal condition (light blue bars) rv0805up showed an insignificant enrichment of PCR signal for PhoP relative to the mock (no antibody control) sample, suggesting low-affinity DNA binding of PhoP under normal conditions. However, IP samples from cells grown under acidic pH showed a significantly higher enrichment of PhoP at the *rv0805* promoter (rv0805up; compare *light blue* bars with *dark blue* bars). As controls, promoter of *msl3* (msl3up), which is controlled by PhoP, and nonspecific 16S rRNAup showed comparable enrichment, and no enrichment under normal and acidic conditions, respectively. Thus, ChIP data showing PhoP recruitment under acidic pH conditions is in agreement with low pH-specific impact of PhoP on *rv0805* expression (*Figure 2B*). Note that PhoP binding to msl3up was used as a positive control.

To examine DNA binding in vitro, we first probed for the PhoP binding site within the upstream regulatory region of *rv0805* (rv0805up) using MEME Bioinformatic software and the consensus PhoP binding sequence (*He and Wang, 2014*). Our results suggest that the likely PhoP binding sequence spans from –127 to –110 (relative to the ORF start site) of rv0805up (*Figure 2—figure supplement 1A*) (p=0.000726). Next, recombinant PhoP was phosphorylated by acetyl phosphate (AcP) and used in EMSA experiments as described earlier (*Pathak et al., 2010*). Consistent with the presence of a PhoP binding site, EMSA results demonstrate that P~PhoP binds to radio-labeled rv0805up to form a complex stable to gel electrophoresis (*Figure 2—figure supplement 1B*).

## Probing PhoP-dependent regulation of *M. tuberculosis rv0805*

To examine whether the regulatory effect was attributable to PhoP activation via phosphorylation, we next grew *phoPR*-KO complemented with either *phoP* (*Figure 3A*) or the entire *phoPR* encoding ORFs (*Figure 3B*), both under normal (pH 7.0; empty bars) and acidic (pH 4.5; black bars) conditions

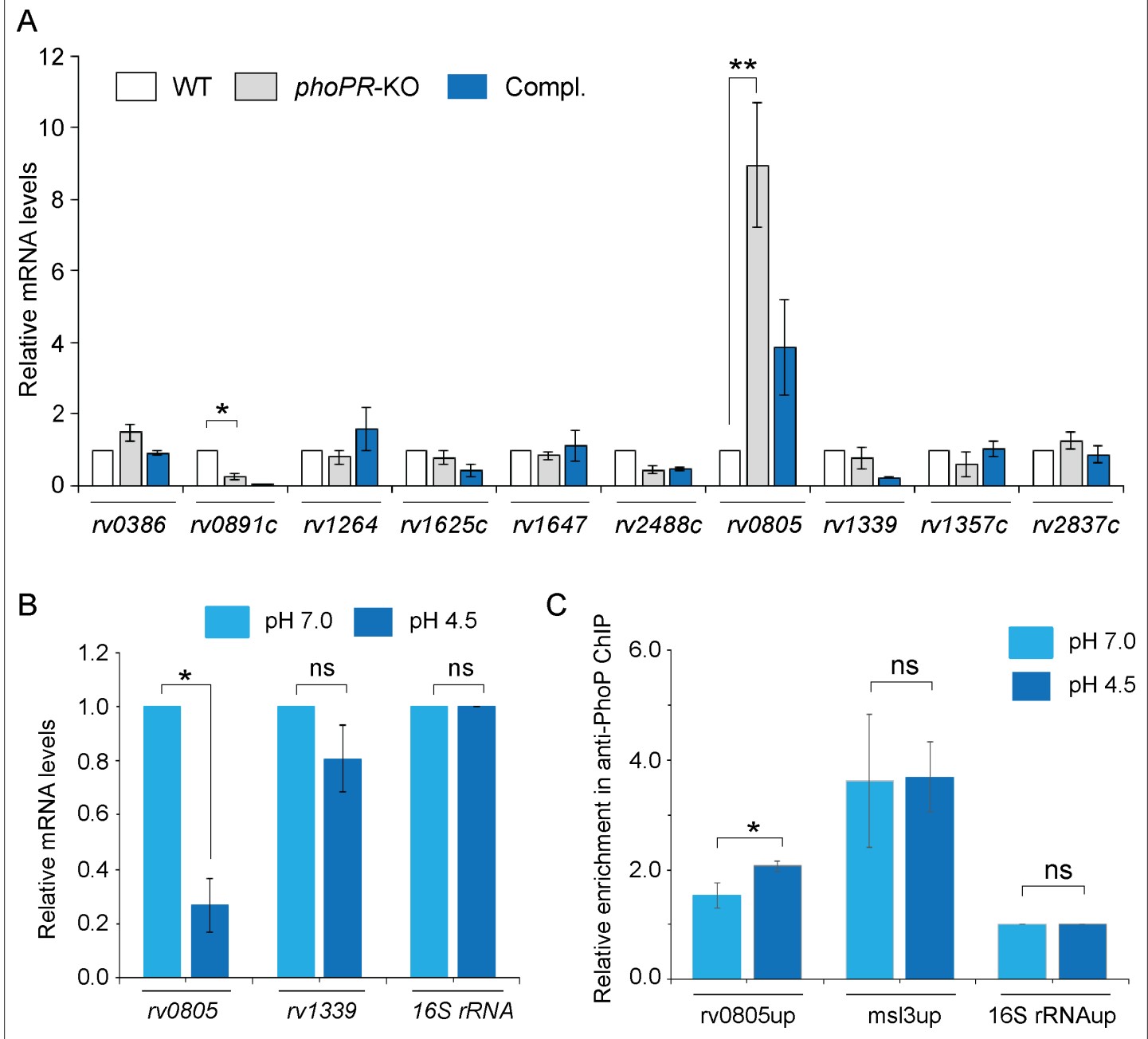

**Figure 2.** PhoP regulates the expression of phosphodiesterase (PDE) *rv0805*. (**A**) To investigate the regulation of 3',5-cyclic adenosine monophosphate (cAMP) level, mRNA levels of well-characterized adenylate cyclases, and phosphodiesterases (PDEs) were compared in indicated mycobacterial strains by RT-qPCR as described in 'Materials and methods'. The results show average values from biological triplicates, each with two technical repeats (*p<0.05; **p<0.01). Note that the difference in expression levels of *rv0805* between WT and *phoPR*-KO was significant (p<0.01), whereas the fold difference in mRNA level between WT and the complemented mutant (Compl.) remains nonsignificant (not indicated). (**B**) To determine the effect of acidic pH conditions of growth, mycobacterial *rv0805* expression was compared in WT grown under normal (pH 7.0) and acidic pH (pH 4.5). Average fold difference in mRNA levels from biological duplicates (each with a technical repeat) was measured as described in 'Materials and methods' (**p≤0.05). As controls, expression of *rv1339* and *16S rDNA* was also measured. Nonsignificant difference is not indicated. (**C**) In vivo PhoP binding to *rv0805* promoter (rv0805up) was compared in WT grown under normal and acidic conditions of growth using anti-PhoP antibody followed by ChIP-qPCR. Fold enrichment data represent mean values of two independent experiments with a statistically significant fold difference (**p-value<0.01; *p-value<0.05). The upstream regulatory regions of 16S rRNA (16S rRNAup) and msl3 (msl3up) were used as negative and positive controls, respectively. The assay conditions, sample analyses, and detection are described in 'Materials and methods'.

The online version of this article includes the following source data and figure supplement(s) for figure 2:

*Figure 2 continued on next page*

*Figure 2 continued*

**Source data 1.** Relative mRNA levels of genes in WT, phoPR mutant and the complemented mutant.

**Source data 2.** Relative mRNA levels of indicated genes under normal and acidic conditions.

**Source data 3.** Relative enrichment of anti-PhoP ChIP within target promoters.

**Figure supplement 1.** Probing in vitro DNA binding of PhoP to the *rv0805* regulatory region.

**Figure supplement 1—source data 1.** EMSA to examine PhoP binding to pde promoter region.

**Figure supplement 1—source data 2.** EMSA to examine PhoP binding to pde promoter region.

and compared relative expression of *rv0805*. Although both strains expressed *phoP*, the former strain lacked a functional copy of *phoR*, the cognate sensor kinase that phosphorylates PhoP (*Gupta et al., 2006*). *M. tuberculosis* H37Rv lacking a *phoR* gene (*phoPR*-KO::*phoP*) did not show a low pH-dependent repression of *rv0805* expression. However, similar to WT bacilli, we observed a low pH-dependent significant downregulation of *rv0805* expression in *phoPR*-KO::*phoPR* (Compl.). Note that a comparable expression of PDE *rv1339* was observed in both strains regardless of growth conditions. These results indicate that acidic pH-dependent repression of *rv0805* expression in vivo is attributable to *P*~PhoP requiring the presence of PhoR.

To examine the effect of Rv0805 on mycobacterial cAMP level, we next expressed a copy of *rv0805* in WT bacteria (referred to as WT-Rv0805) (*Figure 3C*). *rv0805* ORF was cloned within the multicloning site of pSTki (*Parikh et al., 2013*) between the EcoRI and HindIII sites under the control of $P_{myc1}tetO$ promoter, and expression of *rv0805* under non-inducing condition was verified by determining the mRNA level (*Figure 3—figure supplement 1A*). Although copy number of episomal vectors with pAl5000 origin of replication (*oriM*) has been reported to be 3 by Southern hybridization (*Ranes et al., 1990*), in this case wild-type and mutant Rv0805 proteins were expressed from single-copy chromosomal integrants (*Parikh et al., 2013*). We then assessed the impact of Rv0805 on intra-mycobacterial cAMP level (*Figure 3C*). Consistent with a previous study (*Agarwal et al., 2009*), WT-Rv0805 showed a significant depletion (2.1 ± 0.7-fold) of intra-mycobacterial cAMP relative to WT bacteria. To confirm that the reduced level of mycobacterial cAMP is attributable to *rv0805* expression, we also expressed *rv0805M*, a mutant Rv0805 lacking phosphodiesterase activity. As structural data, coupled with biochemical evidence, reveals that Asn-97 at the enzyme active site plays a key role in phosphodiesterase activity of Rv0805 (*Shenoy et al., 2007*; *Shenoy et al., 2005*), the mutant Rv0805M was constructed by changing the conserved Asn-97 to Ala. WT-Rv0805M showed an insignificant variation of cAMP level relative to WT, suggesting that depletion of intra-mycobacterial cAMP in WT-Rv0805 is indeed attributable to phosphodiesterase activity of Rv0805. The corresponding mRNA levels of PDEs (wild-type and the mutant) are overexpressed approximately 4.5- to 6-fold relative to the genomic *rv0805* level of WT-H37Rv (*Figure 3—figure supplement 1A*). In contrast, other PDE encoding genes (*rv1357* and *rv2387*), under identical conditions, demonstrate comparable expression levels in WT-H37Rv and *rv0805* overexpressing strains. Overexpression of these PDEs did not influence bacterial growth under normal conditions (*Figure 3—figure supplement 1B*).

To further probe the regulation of Rv0805 expression and its control of intra-mycobacterial cAMP level, we utilized a previously reported CRISPRi-based approach (*Singh et al., 2016*) to construct *rv0805* and *phoP* knockdown (*rv0805*-KD and *phoP*-KD, respectively) mutants. Consistent with *phoPR*-KO, *phoP*-KD shows a significantly higher *rv0805* expression in the presence of ATc relative to its absence (*Figure 3D*). However, despite a significant downregulation of *rv0805* expression in the presence of ATc, a comparable *phoP* expression was observed in *rv0805*-KD mutant both in the absence or presence of ATc. As a control, we observed a comparable expression of 16S rRNA in both knockdown mutants. Next, we determined intra-mycobacterial cAMP of the mutants as described in *Figure 1* (*Figure 3E*). cAMP level of *phoP*-KD (showing activation of Rv0805) was significantly lower relative to WT bacteria. In contrast, *rv0805*-KD mutant demonstrated a significantly higher level of cAMP relative to WT. We speculate that effective knocking down of *phoP* or *rv0805* is not truly reflected in the extent of variation of cAMP levels possibly due to the presence of numerous other mycobacterial PDEs. These data represent an integrated view of our results, suggesting that PhoP-dependant repression of *rv0805* regulates intra-mycobacterial cAMP level. In keeping with these results, activated PhoP under acidic pH conditions significantly represses *rv0805*, and intracellular

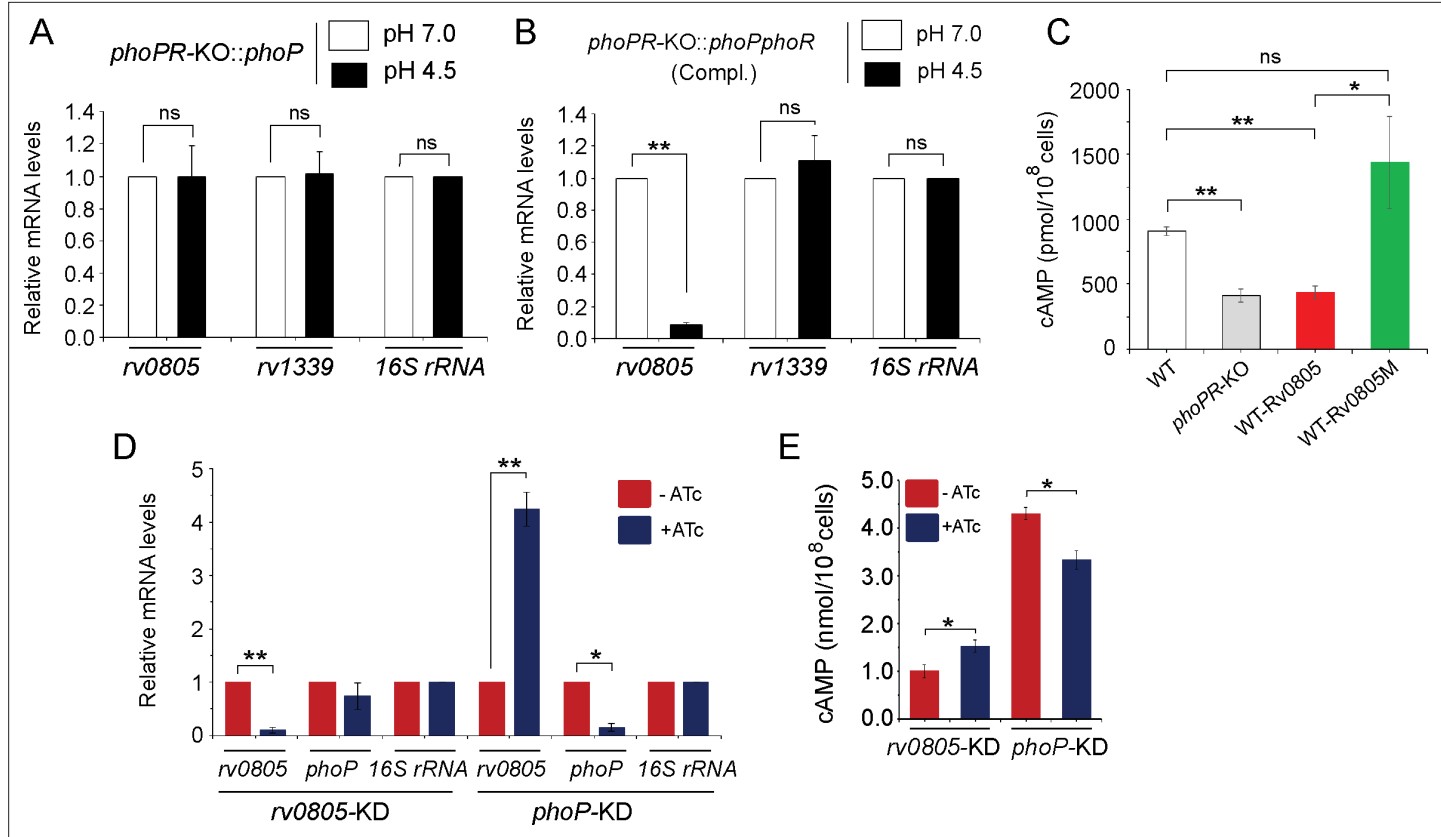

**Figure 3.** PhoP-dependent repression of *rv0805* to maintain mycobacterial 3',5-cyclic adenosine monophosphate (cAMP) level requires the presence of PhoR. (**A, B**) To determine the impact of PhoR (the cognate sensor kinase of PhoP), expression of *rv0805* was compared in indicated *M. tuberculosis* H37Rv strains: (**A**) *phoPR*-KO::*phoP* (*phoPR* mutant complemented with *phoP*) and (**B**) *phoPR*-KO::*phoPR* (*phoPR* mutant complemented with *phoP-phoR*) under normal and acidic conditions of growth. As expected, *phoPR*-KO::*phoPR* (Compl.) shows a significant repression of *rv0805* (but not *rv1339*) under acidic pH (***p<0.001). However, *rv0805* expression remains comparable in *phoPR*-KO::*phoP* under normal as well as acidic conditions of growth. As a control, *rv1339* fails to show a variable expression in indicated mycobacterial strains. (**C**) To determine the effect of ectopic expression of *rv0805* on intra-mycobacterial cAMP level, WT and mutant Rv0805 proteins (Rv0805M, defective for phosphodiesterase activity) were expressed in *M. tuberculosis* H37Rv (to construct WT-Rv0805, and WT-Rv0805M, respectively) as described in 'Materials and methods'. Similar to *phoPR*-KO, WT-Rv0805 (but not WT-Rv0805M) showed a considerably lower level of cAMP relative to WT bacteria. Significance in variation of cAMP levels was determined by paired Student's *t*-test (*p<0.05; **p<0.01). (**D, E**) Relative expression of *phoP* and PDE in *phoP*-KD and *rv0805*-KD (*phoP* and *rv0805* knockdown constructs, respectively). In keeping with elevated expression of *rv0805* in *phoPR*-KO, *phoP*-KD shows an elevated expression of *rv0805* relative to WT bacilli. In contrast, *phoP* expression level remains unaffected in *rv0805*-KD mutant. Panel (**E**) measured corresponding intra-bacterial cAMP levels in the respective knockdown mutants, as described in the legend to *Figure 1A*.

The online version of this article includes the following source data and figure supplement(s) for figure 3:

**Source data 1.** Relative mRNA levels of genes in indicated strains under normal and acidic pH.

**Source data 2.** Relative mRNA levels of genes in indicated strains under normal and acidic pH.

**Source data 3.** Intrabacterial cAMP levels in indicated mycobacterial strains.

**Source data 4.** Relative mRNA levels of genes in indicated knock-down mutants of *M. tuberculosis*.

**Source data 5.** cAMP levels of indicated knock-down mutants.

**Figure supplement 1.** Ectopic expression of wild-type (WT) and mutant *rv0805* (Rv0805M) in WT bacilli.

**Figure supplement 1—source data 1.** Relative mRNA levels of genes in indicated mycobacterial strains.

**Figure supplement 1—source data 2.** CFU enumeration of indicated mycobacterial strains.

mycobacteria most likely utilize a higher level of cAMP to effectively mitigate stress for survival under hostile environment including acidic pH of the phagosome.

## PhoP contributes to mycobacterial stress tolerance by repressing the *rv0805* PDE expression

To investigate whether cAMP level influences mycobacterial susceptibility to stress, we compared in vitro growth under acidic pH (pH 4.5) (*Figure 4A*). As expected, *phoPR*-KO showed a significant growth inhibition relative to WT under low pH (pH 4.5) (*Bansal et al., 2017*). WT-Rv0805, but not WT-Rv0805M, displayed a comparable susceptibility to acidic pH as that of *phoPR*-KO. However, all four mycobacterial strains showed comparable growth at pH 7.0. Next, to compare the growth of WT-Rv0805 and WT under oxidative stress, cells were grown in the presence of increasing concentrations of diamide, a thiol-specific oxidant, and examined by microplate-based Alamar Blue assays (*Figure 4—figure supplement 1A*). We recently showed that *phoPR*-KO is significantly more sensitive to diamide relative to WT (*Goar et al., 2022*). Here, we uncovered that similar to *phoPR*-KO, WT-Rv0805, but not WT-Rv0805M, was significantly more susceptible to diamide stress compared to WT (*Figure 4B*). A previous study reported that *phoP*-deleted mutant strain was more sensitive to cumene hydrogen peroxide (CHP), suggesting a role of PhoP in regulating mycobacterial stress response to oxidative stress (*Walters et al., 2006*). To compare sensitivity to CHP, we grew mycobacterial strains in the presence of 50 μM CHP for 24 hr and determined their survival by enumerating CFU values (*Figure 4C*). In this case, we were unable to perform Alamar Blue-based survival assays requiring a longer time because of the bactericidal property of CHP. Our CFU data highlight that WT-Rv0805, but not WT-Rv0805M, displayed a significantly higher growth inhibition relative to WT in the presence of CHP. Together, these results reveal similar behavior of *phoPR*-KO and WT-Rv0805 by demonstrating a comparably higher susceptibility of these strains to acidic pH and oxidative stress relative to WT bacteria and indicate a link between intra-mycobacterial cAMP level and bacterial stress response. It appears that at least one of the mechanisms by which PhoP contributes to global stress response is attributable to the maintenance of cAMP level.

A previous study showed that *rv0805* overexpression in *M. smegmatis* influences cell wall permeability (*Podobnik et al., 2009*). Having shown a significantly higher sensitivity of WT-Rv0805 to low pH and oxidative stress (relative to WT), we sought to investigate whether altered cell wall structure/ properties of the mycobacterial strain contribute to elevated stress sensitivity. We compared expression level of lipid biosynthetic genes, which encode part of cell wall structure of the bacilli (*Gonzalo Asensio et al., 2006*; *Walters et al., 2006*). Our results suggest that in contrast to *phoPR*-KO, both WT-Rv0805 and WT-Rv0805M share a comparable expression profile of complex lipid biosynthesis genes as that of WT (*Figure 4—figure supplement 1B*). These results suggest that both strains expressing wild-type or mutant PDEs share a largely similar cell wall properties and are consistent with (a) a recent study reporting no significant effect of cAMP dysregulation on mycobacterial cell wall structure/permeability (*Wong et al., 2023*), and (b) role of PhoP in cell wall composition and complex lipid biosynthesis (*Gonzalo Asensio et al., 2006*; *Goyal et al., 2011*; *Walters et al., 2006*). These results support our view that higher susceptibility of WT-Rv0805 to stress conditions is attributable to its reduced cAMP level.

To investigate the impact of mycobacterial cAMP level in vivo, we studied the infection of murine macrophages using WT, WT-Rv0805, and WT-Rv0805M (*Figure 4D*). In this assay, WT-H37Rv inhibits phagosome maturation, whereas phagosomes with *phoPR*-KO mature into phagolysosomes (*Anil Kumar et al., 2016*). In our present experimental setup, although WT bacilli inhibited phagosome maturation, infection of macrophages with WT-Rv0805 and *phoPR*-KO matured into phagolysosomes, suggesting increased trafficking of the bacilli to lysosomes. Under identical conditions, WT-Rv0805M could effectively inhibit phagosome maturation just as WT bacteria. Results from co-localization experiments are plotted in *Figure 4E* and Pearson's correlation coefficient of the quantified co-localization signals in *Figure 4F*. These data suggest reduced ability of WT-Rv0805, but not WT-Rv0805M (relative to WT), to inhibit phagosome maturation. From these results, we suggest that ectopic expression of *rv0805* impacts phagosome maturation, arguing in favor of a role of PhoP in influencing phagosome–lysosome fusion in macrophages.

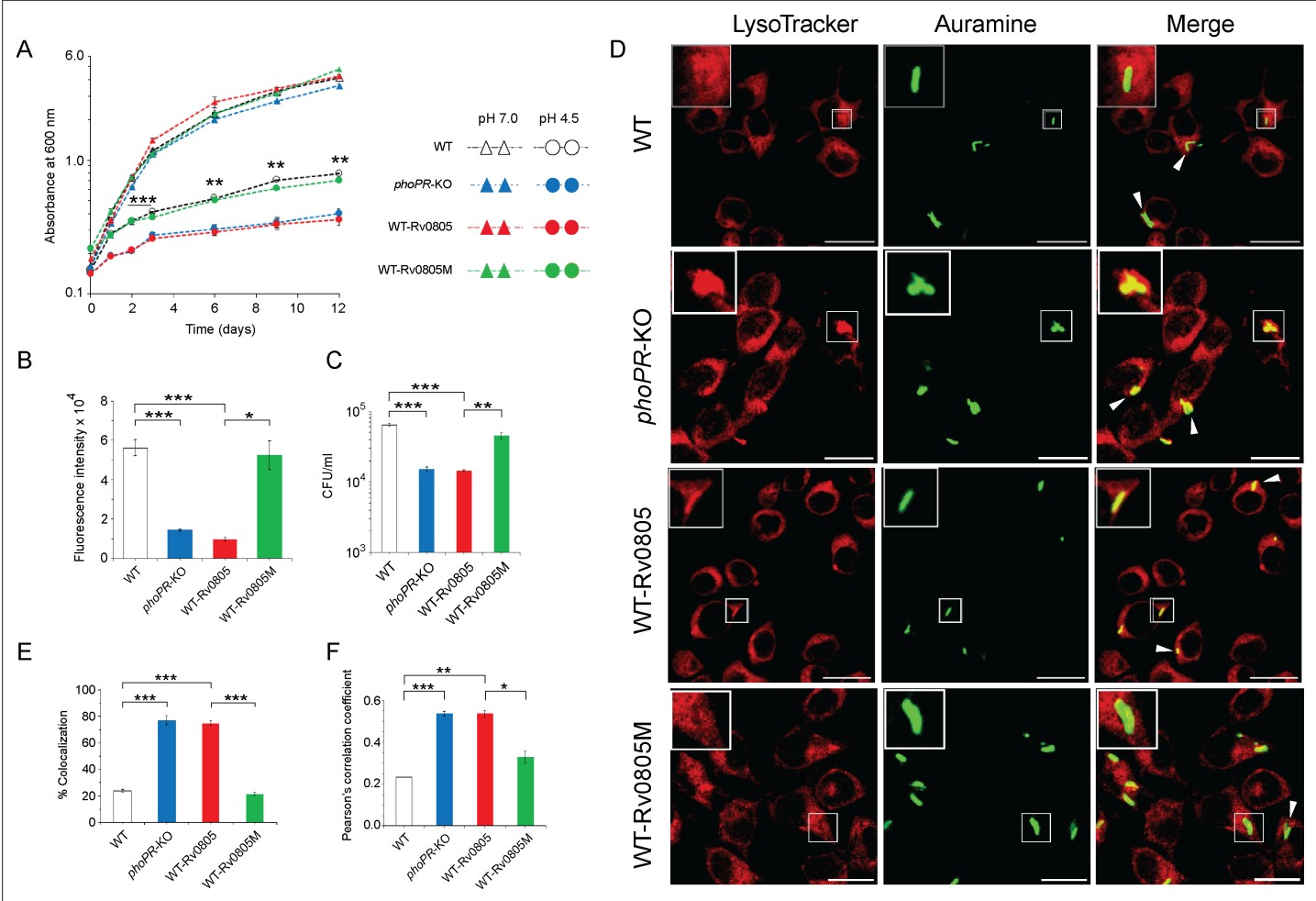

**Figure 4.** Regulation of 3′,5-cyclic adenosine monophosphate (cAMP) level and its effect on mycobacterial stress tolerance and survival in macrophages. (**A**) To compare susceptibility to low pH conditions, the indicated mycobacterial strains were grown at pH 4.5, and similar to *phoPR*-KO (gray circles), WT-Rv0805 (red circles) shows a significant growth defect relative to WT (empty circles). However, WT-Rv0805M (green circles) grows comparably well as that of the WT (empty circles). In contrast, at pH 7.0 all four mycobacterial strains (WT, empty triangles; *phoPR*-KO, gray triangles; WT-Rv0805, red triangles; WT-Rv0805M, green triangles) displayed comparable growth. (**B**) Microplate-based assay using Alamar Blue was utilized to examine mycobacterial sensitivity to increasing concentrations of diamide. In this assay, reduction of Alamar Blue correlates with the change of a non-fluorescent blue to a fluorescent pink appearance, which is directly proportional to bacterial growth. Survival of indicated mycobacterial strains, under normal conditions and in the presence of 5 mM diamide, was determined by plotting fluorescence intensity (*$p<0.05$; **$p<0.01$; ***$p<0.001$). The data is normalized relative to WT grown in the presence of 5 mM diamide. (**C**) To compare susceptibility to stress conditions, these mycobacterial strains were next grown in the presence of 50 µM cumene hydrogen peroxide (CHP). In the presence of CHP, WT-Rv0805 (red column), but not WT-Rv0805M (green column), shows a significant growth defect (relative to WT [empty column]) in striking similarity to *phoPR*-KO (gray column). Note that similar to *phoPR*-KO, WT-Rv0805 shows a comparably higher sensitivity to CHP relative to WT bacilli. However, WT-Rv0805M expressing a mutant Rv0805 shows a significantly lower sensitivity to CHP relative to WT-Rv0805, as measured by the corresponding CFU values. The growth experiments were performed in biological duplicates, each with two technical replicates (**$p≤0.01$; ***$p≤0.001$). (**D**) Murine macrophages were infected with indicated *M. tuberculosis* H37Rv strains. The cellular organelle was made visible by LysoTracker; mycobacterial strains were stained with phenolic auramine solution, and the confocal images display merge of two fluorescence signals (LysoTracker: red; H37Rv: green; scale bar: 10 µm). The insets in the merge panels indicate bacteria, which either have inhibited or facilitated trafficking into lysosomes. White arrowheads in the merge panels indicate non-co-localization, which remains higher in WT-H37Rv and WT-Rv0805M relative to *phoPR*-KO or WT-Rv0805. (**E**) Bacterial co-localization of *M. tuberculosis* H37Rv strains. The percentage of auramine-labeled strains co-localized with LysoTracker was determined by counting at least 100 infected cells in 10 different fields. The results show the average values with standard deviation determined from three independent experiments (***$p≤0.001$). (**F**) Pearson's correlation coefficient of images of internalized auramine-labeled mycobacteria and LysoTracker red marker in RAW 264.7 macrophages. Data are representative of mean ± SD, derived from three independent experiments (*$p<0.05$; ***$p<0.001$).

The online version of this article includes the following source data and figure supplement(s) for figure 4:

**Source data 1.** Growth curves of indicated strains under normal and acidic pH.

*Figure 4 continued on next page*

*Figure 4 continued*

**Source data 2.** Relative flourescence intensity of indicated mycobacterial strains (Alamar Blue assays).

**Source data 3.** CFU of indicated strains under oxidative stress.

**Source data 4.** Confocal images showing colocalization or lack thereof of indicated mycobacterial strains in macrophage infection studies.

**Source data 5.** % Colocalization of indicated mycobacterial strains in infection studies.

**Source data 6.** Pearson's correlation coefficient of indicated strains in infection studies.

**Figure supplement 1.** PhoP-dependent *rv0805* expression contributes to mycobacterial survival under oxidative stress.

**Figure supplement 1—source data 1.** Alamar Blue assay of indicated strains (unlabeled data).

**Figure supplement 1—source data 2.** Alamar Blue assay of indicated strains in the presence of diamide.

**Figure supplement 1—source data 3.** Relative mRNA levels of specific genes in indicated mycobacterial strains.

## Intra-bacterial cAMP level and its effect on in vivo survival of mycobacteria

To examine the effect of intra-bacterial cAMP level on in vivo survival, mice were infected with mycobacterial strains via the aerosol route. Day 1 post-infection, CFU analyses revealed a comparable count of four mycobacterial strains (~100 bacilli) in the mice lungs. However, for WT-Rv0805, the CFU recovered from infected lungs 4 wk post-infection declined by ~218-fold relative to the lungs infected with WT bacteria (*Figure 5A*). In contrast, the CFU recovered from infected lungs after 4 wk of infection by WT-Rv0805M marginally declined by approximately sevenfold relative to the lungs infected with WT bacilli. These results suggest a significantly compromised ability of WT-Rv0805 (relative to WT) to replicate in the mice lungs. Note that *phoPR*-KO, under the conditions examined, showed an ~246-fold lower lung burden compared to WT. In keeping with these results, while the WT bacilli disseminated to the spleens of infected mice, a significantly lower count of WT-Rv0805 was recovered from the spleens after 4 wk of infection (*Figure 5B*). Thus, we suggest that one of the reasons that accounts for an attenuated phenotype of *phoPR*-KO in both cellular and animal models is attributable to PhoP-dependent repression of *rv0805* PDE activity, which controls mycobacterial cAMP level.

*M. tuberculosis* H37Rv persists within granulomas where it is protected from the anti-mycobacterial immune effectors of the host. Histopathological evaluations showed that WT bacilli-infected lung sections displayed aggregation of granulocytes within alveolar spaces that degenerate progressively to necrotic cellular debris. In contrast, *phoPR*-KO and WT-Rv0805 showed less severe pathology as indicated by decreased tissue consolidation, smaller granulomas, and open alveolar space (*Figure 5C*). Together, these results suggest that ectopic expression of *rv0805* in WT bacilli is phenotypically equivalent to deletion of *phoP*, suggesting that failure to maintain cAMP level most likely accounts for attenuated phenotype of the bacilli and absence of immunopathology in the lungs of infected mice.

## Discussion

A number of studies suggest that conditions associated with host environment like low pH and macrophage interactions often influence mycobacterial cAMP levels (*Bai et al., 2009*; *Gazdik and McDonough, 2005*). Although many bacterial pathogens modulate host cell cAMP levels as a common strategy, the mechanism of regulation of mycobacterial cAMP-level remains unknown. In this study, we sought to investigate whether PhoP, a master regulator implicated in controlling diverse mycobacterial stress response, regulates mycobacterial cAMP level. We find that under normal conditions as well as under carefully controlled single stress conditions, *phoPR*-KO shows a significantly lower level of cAMP relative to the WT bacilli (*Figure 1A*), and complementation of the mutant restored cAMP level. To investigate the mechanism, we next probed regulation of ACs and PDEs (*Figure 2*) and demonstrated that PhoP functions as a major repressor of *rv0805*, encoding cAMP-specific PDE. Indeed, this newly identified *rv0805* regulation, coupled with a recent discovery that phosphodiesterase activity of Rv0805 controls propionate detoxification (*McDowell et al., 2023*), fits well with and explains the previously puzzling in vivo observation by *Abramovitch et al., 2011* that PhoP-controlled *aprABC* locus is associated with the regulation of genes of carbon and propionate metabolism.

Although a large number of ACs are present in *M. tuberculosis* genome, a class III metallo-phosphoesterase Rv0805 was earlier considered the only PDE, specific for mycobacterial cAMP.

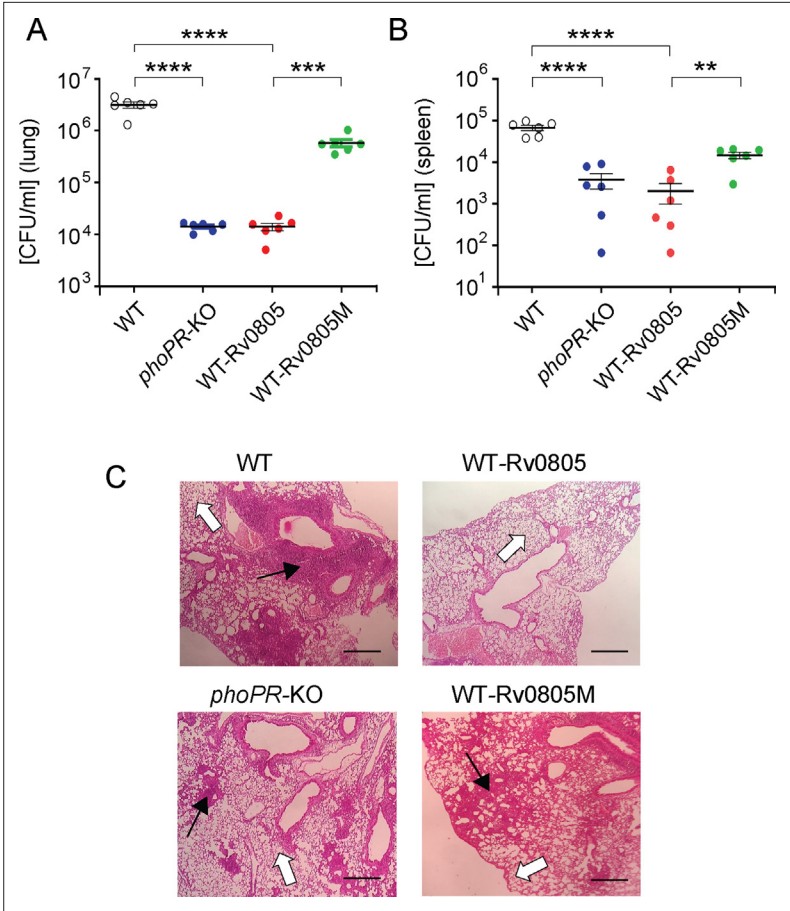

**Figure 5.** Dysregulation of mycobacterial 3′,5-cyclic adenosine monophosphate (cAMP) level impacts mycobacterial survival in vivo. (**A, B**) Survival of mycobacterial strains in mice (**A**) lung and (**B**) spleen after animals were given an aerosol infection with ~100 CFU/lung. Mycobacterial load represents mean CFU values with standard deviations obtained from at least five animals per strains used (**p<0.01; ***p<0.001; ****p<0.0001). (**C**) Histopathology of lung sections after 4 wk of infection with indicated bacterial strains. Sections were stained with hematoxylin and eosin, observed under a light microscope, and images of the pathology sections collected at ×40 magnification display granulomas (filled arrows) and alveolar space (empty arrows) (scale bar, 200 μm).

The online version of this article includes the following source data for figure 5:

**Source data 1.** CFU of indicated mycobacterial strains in mice lung (lung burden).

**Source data 2.** CFU of indicated mycobacterial strains in mice spleen (spleen burden).

**Source data 3.** Histopathology of lung section unlabeled.

**Source data 4.** Histopathology of lung section of indicated strains.

However, a recent study has identified an atypical class II PDE Rv1339, which upon overexpression reduces cAMP level and contributes to antibiotic sensitivity (*Thomson et al., 2022*). While the functional role of Rv1339 in *M. tuberculosis* is yet to be understood, crystal structure and biochemical evidence suggest that dimeric Rv0805 is stabilized by the presence of a divalent cation and remains catalytically active on a broad range of linear and cyclic PDE substrates in vitro (*Keppetipola and Shuman, 2008*; *Shenoy et al., 2007*; *Shenoy et al., 2005*). More recently, cyclic nucleotide hydrolytic activity of mycobacterial Rv0805 has been implicated in propionate detoxification (*McDowell et al., 2023*). However, the mechanism of regulation of Rv0805 and its effect on mycobacterial cAMP level remained unknown before the present study.

To examine biological significance of PhoP-dependent Rv0805 expression, we studied *rv0805* expression under acidic conditions of growth as *phoPR* system is induced under acidic pH both in vitro and in macrophages (*Abramovitch et al., 2011*; *Bansal et al., 2017*). A significantly higher repression of *rv0805* expression under acidic pH relative to normal conditions is consistent with the

activation of PhoP and subsequent repression of *rv0805* (*Figure 3*). These results further suggest that effective mitigation of stress by mycobacteria possibly requires a higher cAMP level for survival under intra-phagosomal environment. In keeping with these results, we find that (a) *P~PhoP* binds to *rv0805* regulatory region (*Figure 2—figure supplement 1*) and (b) PhoP-dependent *rv0805* expression requires PhoR (*Figure 3A and B*), the cognate kinase which activates PhoP in a signal-dependent manner (*Gupta et al., 2006*; *Singh et al., 2023*). These results account for a consistently lower level of cAMP in *phoPR*-KO relative to the WT bacilli. Notably, except recently reported PDE Rv1339, Rv0805 has been known as the only cAMP-specific PDE present in the slow-growing pathogenic mycobacteria and its closely related species (*Matange, 2015*; *Shenoy et al., 2007*; *Shenoy et al., 2005*), and Rv1339 expression does not appear to be regulated by the *phoP* locus (*Figure 2*). Thus, the above results showing PhoP-dependent repression of *rv0805* activity likely represent the most critical step of regulation of mycobacterial cAMP level under stress. In this connection, our recent results that PhoP interacts with cAMP receptor protein, CRP, and a complex of two interacting regulators control expression of virulence determinants (*Khan et al., 2022*) invite speculation of a complex regulatory control of cAMP-responsive mycobacterial physiology.

As one might argue that PhoP deletion and *rv0805* overexpression could be unrelated and independent events, we constructed *phoP* and *rv0805* knockdown mutants to further investigate the PhoP-Rv0805-cAMP pathway. Our objective was to probe regulation of expression (*Figure 3D*) and examine the impact on mycobacterial cAMP (*Figure 3E*). *phoP*-KD significantly elevated *rv0805* expression; however, *phoP* expression remains unaffected in *rv0805*-KD (*Figure 3D*). While elevated *rv0805* expression in *phoP*-KD reduces cAMP level, understandably cAMP level is elevated in *rv0805*-KD mutant (*Figure 3E*). These results integrate PhoP-dependent *rv0805* repression with mycobacterial cAMP level, suggesting how *phoP*-deletion or knockdown, at least in part, mimics Rv0805 overexpression. These considerations take on more significance given the fact that these two events have similar consequences on relevant strains with respect to stress tolerance and survival in cellular and animal models. Thus, our results suggest that ectopic expression of *rv0805* is functionally equivalent to deletion of the *phoP* locus. This observation is in apparent conflict with a previous work by *Matange*

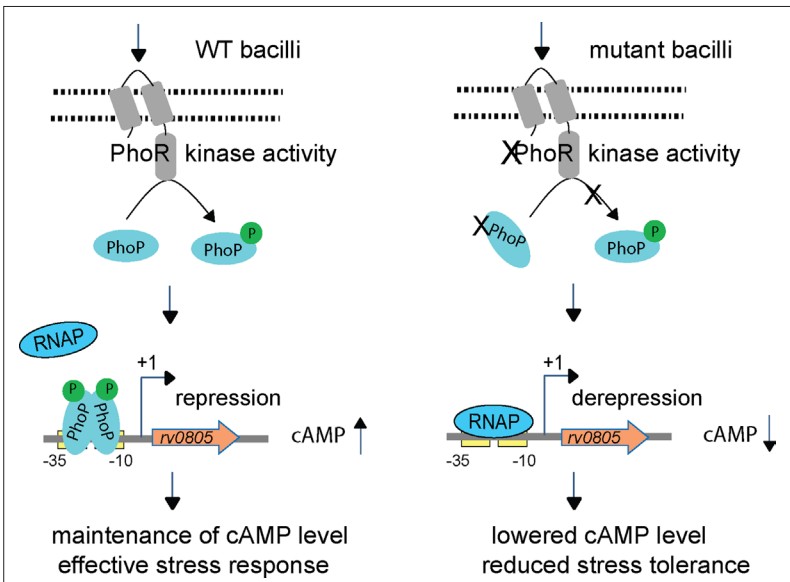

**Figure 6.** Increased 3',5-cyclic adenosine monophosphate (cAMP) level and effective stress response versus decreased cAMP level and reduced stress tolerance of mycobacteria. In this model, upon activation by an appropriate signal via the cognate sensor PhoR, *P~PhoP* binds to *rv0805* regulatory region and functions as a specific repressor by preventing access for mycobacterial RNA polymerase (RNAP) to bind to the promoter and initiate transcription. In keeping with PhoP-dependent *rv0805* repression, our results demonstrate a reproducibly lower level of cAMP in *phoPR*-KO relative to WT bacilli. Thus, *phoPR*-KO (or WT-Rv0805) remains ineffective to mount an appropriate stress response most likely due to its inability to coordinate regulated gene expression because of dysregulation of cAMP level, accounting for their reduced stress tolerance. Together, these molecular events suggest that failure to maintain cAMP level accounts for attenuated phenotype of the bacilli.

*et al., 2013*, suggesting a cAMP-independent transcriptional response in *rv0805* overexpressing *M. tuberculosis* H37Rv. Although both studies were performed with *rv0805* overexpressing bacilli, the fact that important differences in the expression of PDEs in this study (*Matange et al., 2013*) and in our assays – yielding significantly different levels of *rv0805* expression – most likely account for this discrepancy. While we cannot completely rule out the possibility of cleavage of other cyclic nucleotide(s) by Rv0805 (*Keppetipola and Shuman, 2008*; *Shenoy et al., 2007*; *Shenoy et al., 2005*) impacting our results, consistent with a previous study, our results correlate *rv0805* expression with intra-mycobacterial cAMP level (*Agarwal et al., 2009*). Further, our data on the effect of expression of cyclic nucleotide-specific PDE Rv0805 or its inactive mutant (Rv0805M) correlate well with enzyme activities of the corresponding PDEs on mycobacterial cAMP levels (*Figure 3C*). Thus, we infer that PhoP-dependent regulation of *rv0805* is a critical regulator of intra-mycobacterial cAMP level.

Our experiments to understand the physiological significance of PhoP-dependent repression of *rv0805* expression uncover a comparable stress tolerance of WT-Rv0805 and *phoPR*-KO (significantly reduced relative to WT) (*Figure 4*). These results are consistent with the notion that cAMP level, at least in part, accounts for mycobacterial stress response. Along the line, WT-Rv0805 displayed a reduced ability to inhibit phagosome-lysosome fusion like *phoPR*-KO (*Figure 4*). Further, we show that WT-Rv0805, unlike the WT bacilli or WT-Rv0805M, shows a significantly reduced intracellular growth in mice as that of *phoPR*-KO (*Figure 5*). Thus, these results are of fundamental significance to establish that PhoP contributes to the maintenance of cAMP level and integrates it to the mechanisms of mycobacterial stress tolerance and intracellular survival. Together, we identify a novel mycobacterial pathway as a therapeutic target and provide yet another example of an intimate link between bacterial physiology and intracellular survival of the tubercle bacilli.

The results reported here are presented schematically in *Figure 6*. In summary, upon sensing low acidic pH as a signal PhoR activates PhoP, P~PhoP binds to *rv0805* upstream regulatory region and functions as a specific repressor of Rv0805. Therefore, we observed (a) a reproducibly lower level of cAMP in *phoPR*-KO relative to WT-H37Rv, (b) a significantly reduced expression of *rv0805* in WT-H37Rv, grown under acidic pH relative to normal conditions, and (c) comparable cAMP levels in *phoPR*-KO and WT-Rv0805. This is why the two strains remain ineffective to mount an appropriate stress response, most likely due to their inability to coordinate regulation of gene expression because of dysregulation of intra-mycobacterial cAMP level. However, without uncoupling regulatory control of PhoPR and *rv0805* expression, we cannot confirm that dysregulation of cAMP level accounts for virulence attenuation of *phoPR*-KO. Given the fact that *rv0805*-depleted *M. tuberculosis* is growth attenuated in vivo (*McDowell et al., 2023*), paradoxically ectopic expression of *rv0805* leads to dysregulated metabolic adaptation, thereby resulting in reduced stress tolerance and intracellular survival.

## Materials and methods
### Bacterial strains and culture conditions

*Escherichia coli* DH5α was used for cloning experiments. *E. coli* BL21(DE3), an *E. coli* B strain lysogenized with λ DE3, a prophage expressing T7 RNA polymerase from the isopropyl-β-D-thiogalactopyranoside (IPTG)-inducible *lac*UV5 promoter (*Studier and Moffatt, 1986*), was used as the host for overexpression of recombinant proteins. *E. coli* strains were grown in LB medium at 37°C with shaking, transformed according to standard procedures, and the transformants were selected on media containing appropriate antibiotics plates. WT- and mutant *M. tuberculosis* were grown at 37°C in Middlebrook 7H9 broth (Difco) containing 0.2% glycerol, 0.05% Tween-80, and 10% albumin–dextrose–catalase (ADC) or on 7H10-agar medium (Difco) containing 0.5% glycerol and 10% oleic acid–albumin–dextrose–catalase (OADC) enrichment. *phoPR* disruption mutant of *M. tuberculosis* H37Rv (*phoPR*-KO, a kind gift of Dr. Issar Smith) was constructed as described (*Walters et al., 2006*). To this end, a kanamycin-resistant cassette from pUC-K4 was inserted into a unique EcoRV site within the coding region of *phoP* gene, and disruption was confirmed by Southern blot analysis of chromosomal DNA isolated from the mutant. Next, purified plasmid DNAs were electroporated into wild-type *M. tuberculosis* strain by standard protocol (*Jacobs et al., 1991*). To complement *phoPR* expression, pSM607 containing a 3.6 kb DNA fragment of *M. tuberculosis* phoPR including 200 bp *phoP* promoter region, a hygromycin resistance cassette, *attP* site, and the gene encoding phage L5 integrase, as detailed earlier (*Walters et al., 2006*), was used to transform *phoPR* mutant

to integrate at the L5 *attB* site. Growth, transformation of wild-type (WT), *phoPR*-KO, the complemented mutant (Compl.) *M. tuberculosis,* and selection of transformants on appropriate antibiotics plates were performed as described (**Goyal et al., 2011**). When appropriate, antibiotics were used at the following concentrations: hygromycin (hyg), 250 µg/ml for *E. coli* or 50 µg/ml for mycobacterial strains; streptomycin (str), 100 µg/ml for *E. coli* or 20 µg/ml for mycobacterial strains; kanamycin (kan), 20 µg/ml for mycobacterial strains. For in vitro growth under specific stress conditions, the indicated mycobacterial strains were grown to the mid-log phase ($OD_{600}$ 0.4–0.6) and exposed to different stress conditions. For acid stress, cells were initially grown in 7H9 media, pH 7.0, and on attaining the mid-log phase it was transferred to acidic media (7H9 media, pH 4.5), and grown for further 2 hr at 37°C. For oxidative stress, cells were grown in the presence of 50 µM CHP (Sigma) for 24 hr or the indicated diamide concentration(s) for 7 d. For NO stress, cells grown to the mid-log phase were exposed to 0.5 mM DetaNonoate for 40 min (**Voskuil et al., 2003**).

## cAMP measurement

Mycobacterial cell pellets were collected and washed with 1× PBS buffer, cells were resuspended in IP buffer (50 mM Tris pH 7.5, 150 mM NaCl, 1 mM EDTA pH 8.0, 1 mM PMSF, 5% glycerol, and 1% TritonX 100) and CL were prepared by lysing the cells in the presence of Lysing Matrix B (100 µm silica beads; MP Bio) using a FastPrep-24 bead beater (MP Bio) at a speed setting of 6.0 for 30 s. The procedure was repeated for 10 cycles with incubation on ice in between pulses. The supernatant was collected by centrifugation at 13,000 rpm for 10 min and filtered through a 0.22 µm filter (Millipore). cAMP levels in the cells were determined in a plate reader by using fluorescence-based cAMP detection kit (Abcam) according to the manufacturer's recommendations and normalized to the total protein present in the samples as determined by a BCA protein estimation kit (Pierce). For secretion studies, each mycobacterial strain was grown in Sauton's media as described (**Anil Kumar et al., 2016**), comparable counts of bacterial cells were pelleted, resuspended in 2 ml of Sauton's media in a 6-well plate format for 2 hr at 37°C, and the supernatants (CFs) were collected for cAMP measurements, as described previously (**Anil Kumar et al., 2016**).

## Cloning

*M. tuberculosis* full-length ORFs of interest were cloned between EcoRI and HindIII sites of the mycobacterial expression vector pSTKi (**Parikh et al., 2013**) and expressed from the $P_{myc1}tetO$ promoter. Mutation in Rv0805 was introduced by two-stage overlap extension method using mutagenic primers (**Supplementary file 1b**), and the construct was verified by DNA sequencing. For overexpression of WT- or mutant PDEs, WT bacilli were transformed with pST-rv0805 or pST-rv0805M to generate WT-Rv0805 or WT-Rv0805M, respectively.

## EMSA

rv0805up DNA probe was PCR-amplified, resolved on an agarose gel, recovered by gel extraction, and end-labeled with [γ-$^{32}$P ATP] (1000 Ci nmol$^{-1}$) using T4 polynucleotide kinase. The end-labeled DNA probe was purified from free label by Sephadex G-50 spin columns (GE Healthcare) and incubated with increasing amounts of purified PhoP in a total volume of 10 µl binding mix (50 mM Tris–HCl, pH 7.5, 50 mM NaCl, 0.2 mg/ml of bovine serum albumin, 10% glycerol, 1 mM dithiothreitol, ≈50 ng of labeled DNA probe, and 0.2 µg of sheared herring sperm DNA) at 20°C for 20 min. DNA–protein complexes were resolved by electrophoresis on a 6% (w/v) polyacrylamide gel (non-denaturing) in 0.5× TBE (89 mM Tris-base, 89 mM boric acid, and 2 mM EDTA) at 70 V and 4°C, and the position of the radioactive material was determined by exposure to a phosphor storage screen.

## Construction of *M. tuberculosis phoP* and *rv0805* knockdown mutants

In this study, we utilized a previously reported CRISPRi system (**Singh et al., 2016**) to construct knockdown mutants of *phoP* and r*v0805* (*phoP*-KD and *rv0805*-KD, respectively). This approach efficiently inhibits expression of target genes via inducible expression of dCas9 along with gene-specific guide RNAs (sgRNA). The RNAs were 20 nt long and complementary to the non-template strand of the target gene. The sgRNAs of *phoP* and *rv0805* were cloned in a vector pRH2521 using BbsI enzyme, and the constructs were confirmed by sequencing. The corresponding clones were used to transform *M. tuberculosis* harboring pRH2502, a vector expressing an inactive version of *Streptococcus*

*pyogenes* cas9 (dcas9). To express dcas9 and repress sgRNA-targeted genes (*phoP* or *rv0805*), the bacterial strains were grown with or without 600 ng/ml of anhydro-tetracycline (ATc) every 48 hr, and cultures were grown for 4 d. RNA isolation was carried out, and RT-qPCR experiments verified significant repression of target genes. For the induced strains (in the presence of ATc) expressing sgRNAs targeting +155 to +175 (relative to *phoP* translational start site) and +224 to +244 sequences (relative to *rv0805* translational start site), we obtained approximately 85 and 90% reduction of *phoP* and *rv0805* RNA abundance, respectively, relative to corresponding uninduced strains. The oligonucleotides used to generate gene-specific sgRNA constructs and the plasmids utilized in knockdown experiments are listed in (*Supplementary file 1b*).

## RNA isolation

Total RNA was extracted from exponentially growing bacterial cultures grown with or without specific stress as described above. Briefly, 25 ml of bacterial culture was grown to the mid-log phase ($OD_{600}$=0.4–0.6) and combined with 40 ml of 5 M guanidinium thiocyanate solution containing 1% β-mercaptoethanol and 0.5% Tween-80. Cells were pelleted by centrifugation and lysed by resuspending in 1 ml Trizol (Ambion) in the presence of Lysing Matrix B (100 µm silica beads; MP Bio) using a FastPrep-24 bead beater (MP Bio) at a speed setting of 6.0 for 30 s. The procedure was repeated for 2–3 cycles with incubation on ice in between pulses. Next, CLs were centrifuged at 13,000 rpm for 10 min; the supernatant was collected and processed for RNA isolation using Direct-Zol RNA isolation kit (ZYMO). Following extraction, RNA was treated with DNAse I (Promega) to degrade contaminating DNA, and integrity was assessed using a Nanodrop (ND-1000, Spectrophotometer). RNA samples were further checked for intactness of 23S and 16S rRNA using formaldehyde-agarose gel electrophoresis, and Qubit fluorometer (Invitrogen).

## Quantitative real-time PCR

cDNA synthesis and PCR reactions were carried out using total RNA extracted from each bacterial culture, and Superscript III platinum-SYBR green one-step qRT-PCR kit (Invitrogen) with appropriate primer pairs (2 µM) using an ABI real-time PCR detection system. The oligonucleotide primer sequences used in RT-qPCR experiments are listed in (*Supplementary file 1a*). Control reactions with platinum Taq DNA polymerase (Invitrogen) confirmed the absence of genomic DNA in all our RNA preparations, and endogenously expressed *M. tuberculosis rpoB* was used as an internal control. Fold difference in gene expression was calculated using the $\Delta\Delta C_T$ method (*Schmittgen and Livak, 2008*). The average fold differences in mRNA levels were determined from at least two biological repeats each with two technical repeats. Nonsignificant difference is not indicated.

## ChIP assays

ChIP experiments in actively growing cultures of *M. tuberculosis* were carried out essentially as described previously (*Fol et al., 2006*). Immunoprecipitation (IP) was performed using anti-PhoP antibody and protein A/G agarose beads (Pierce). qPCR reactions included PAGE purified primer pairs (*Supplementary file 1a*) spanning specific promoter regions using suitable dilutions of IP DNA in a reaction buffer containing SYBR green mix, and one unit of Platinum Taq DNA polymerase (Invitrogen). An IP experiment without adding antibody (mock) was used as the negative control, and data was analyzed relative to PCR signal from the mock sample. PCR amplifications were carried out for 40 cycles using serially diluted DNA samples (mock, IP treated, and total input) in a real-time PCR detection system (Applied Biosystems). In all cases, melting curve analysis confirmed amplification of a single product.

## Immunoblotting

CLs or CFs were resolved by 12% SDS-PAGE and visualized by western blot analysis using appropriate antibodies. Briefly, resolved samples were electroblotted onto polyvinyl difluoride (PVDF) membranes (Millipore, USA) and were detected by anti-GroEL2 antibody (Sigma), anti-CFP-10 antibody (Abcam), or affinity-purified anti-PhoP antibody elicited in rabbit (Alpha Omega Sciences, India). Goat anti-rabbit secondary antibody conjugated to horseradish peroxidase was used, and blots were developed with the Luminata Forte Chemiluminescence reagent (Millipore). RNA polymerase was used as a

loading control and detected with monoclonal antibody against β-subunit of RNA polymerase, RpoB (Abcam).

## Alamar Blue assay

In this assay, reduction of Alamar Blue correlates with the change of a non-fluorescent blue to a fluorescent pink appearance, which is directly linked to bacterial growth. *M. tuberculosis* H37Rv was grown in 7H9 media (Difco) with 10% ADS (albumin, dextrose, and NaCl) to an $OD_{600}$ of 0.4 and freshly diluted to $OD_{600}$ of 0.02. Next, increasing concentrations of diamide were added to the wells of a 96-well plate containing 0.05 ml 7H9 media followed by addition of 0.05 ml of *M. tuberculosis* H37Rv culture (0.02 $OD_{600}$). The plate was incubated at 37°C for 7 d. Finally, 0.02 ml of 0.02% Resazurin (sodium salt, MP Bio), prepared in sterile 7H9 media, was added to each of the wells and the change in color was examined after incubation at 37°C for 16 hr. The fluorescence excitation was at 530 nm and emission was recorded at 590 nm. The efficiency of inhibition was calculated relative to control wells that did not include diamide, and rifampicin was included as a positive control to confirm the validity of the assay.

## Macrophage infections

Virulence of the indicated H37Rv strains was assessed in murine macrophages according to the previously published protocol (*Solans et al., 2014*). RAW264.7 macrophage was from ATCC, the identity was authenticated by ATCC through STR profiling, and mycoplasma contamination was not detectable. Briefly, macrophages grown in DMEM containing 10% fetal bovine serum at 37°C under 5% $CO_2$, and seeded onto #1 thickness, 18-mm-diameter glass coverslips in a 12-well plate at a 40% confluency (0.5 million cells). Cells were independently infected with titrated cultures of WT, WT-Rv0805, WT-Rv0805M, and *phoPR*-KO strains at a multiplicity of infection of 1:5 for 3 hr at 37°C in 5% $CO_2$, followed by 1× PBS washes thrice. The macrophages were further incubated for 3 hr at 37°C. After infection, extracellular bacteria were removed by washing thrice with PBS. To visualize trafficking of the tubercle bacilli, mycobacterial strains were stained with phenolic auramine solution (which selectively binds to mycolic acids) for 15 min followed by washing with acid alcohol solution and finally with 1× PBS. The cells were stained with 150 nM LysoTracker Red DND-99 (Invitrogen) for 30 min in a $CO_2$ incubator. Next, the cells were fixed with 4% paraformaldehyde for 15 min, washed thrice with PBS, the coverslips were mounted in Slow Fade-Anti-Fade (Invitrogen), and analyzed using laser scanning confocal microscope (Nikon) equipped with Argon (488 nm excitation line; 510 nm emission detection) and LD (561 nm excitation line; 594 nm emission detection) laser lines. Digital images were processed with IMARIS imaging software (version 9.20). Details of the experimental methods and the laser/detector settings were optimized using macrophage cells infected with WT-H37Rv as described previously (*Anil Kumar et al., 2016*). A standard set of intensity threshold was made applicable for all images, and percent bacterial co-localization was determined by analyses of at least 50 infected cells originating from 10 different fields of each of the three independent biological repeats.

## Mouse infections

All experiments pertaining to mice were in accordance with the institutional regulations after a review of the protocols and approval by the Institutional Animal Ethics Committee (IAEC/17/05 and IAEC/19/02). Mice were maintained and bred in the animal house facility of CSIR-IMTECH. Animal infection studies and subsequent experiments were carried out in the Institutional BSL-3 facility as per institutional biosafety guidelines. Briefly, the experiments were conducted with 8- to 10-week-old C57BL/6 mice, infected intranasally, and euthanized post-infection for the evaluation of bacterial load in the lungs and spleens. Infections were given through the respiratory route using an inhalation exposure system (Glass-col) with passaged *M. tuberculosis* H37Rv cultures of the mid-log phase. The actual bacterial load delivered to the animals was enumerated from 3 to 5 aerogenically challenged mice, 1 d post-aerosol challenge. The animals were found to achieve a bacillary deposition of 100–200 CFU in the lungs for each strain. Four weeks post infection, the animals were sacrificed by cervical dislocation, lungs and spleens were isolated aseptically from the euthanized animals, homogenized in sterile 1× PBS, and plated after serially diluting the lysates on 7H11 agar plates, supplemented with 10% OADC and antibiotics (50 μg/ml carbenicillin, 30 μg/ml polymyxin B, 10 μg/ml vancomycin, 20 μg/ml trimethoprim, 20 μg/ml cycloheximide, and 20 μg/ml amphotericin B) to enumerate CFU.

For histopathology, left lung lobes were fixed in 10% buffered formalin, embedded in paraffin, and stained with hematoxylin and eosin for visualization under the microscope. The level of pathology was scored by analyzing perivascular cuffing, leukocyte infiltration, multinucleated giant cell formation, and epithelial cell injury.

## Statistical analysis

Data are presented as arithmetic means of the results obtained from multiple replicate experiments ± standard deviations. Statistical significance was determined by Student's paired $t$-test using Microsoft Excel or GraphPad Prism. Statistical significance was considered at p-values of 0.05 or lower (*$P \leq 0.05$; **$P \leq 0.01$; ***$P \leq 0.001$; ****$P \leq 0.0001$).

# Acknowledgements

We are grateful to G Marcela Rodriguez and Issar Smith (PHRI, New Jersey Medical School, UMDNJ) for ΔphoP-H37Rv, and the complemented *M. tuberculosis* H37Rv strains, Adrie Steyn (University of Alabama) for pUAB300/pUAB400 plasmids, Ashwani Kumar for very helpful discussions, and Sanjeev Khosla for critical reading of the manuscript. We thank the members of the Institutional Animal Facility (iCARE) for their help with approval of our projects from the Institutional Animal Ethics Committee. This study received financial support from intramural grants of CSIR-IMTECH (OLP-0170), CSIR (MLP-0049), and a research grant (to DS) from SERB (EMR/2016/004904), Department of Science and Technology (DST). HK, PP, HG, BB, and NB were supported by CSIR pre-doctoral fellowships.

# Additional information

## Funding

| Funder | Grant reference number | Author |
| --- | --- | --- |
| Institute of Microbial Technology | OLP-0170 | Dibyendu Sarkar |
| Council of Scientific & Industrial Research | MLP-0049 | Dibyendu Sarkar |
| Science and Engineering Research Board | EMR/2016/004904 | Dibyendu Sarkar |
| Council of Scientific & Industrial Research | Pre-doctoral fellowship | Hina Khan<br>Partha Paul<br>Harsh Goar<br>Bhanwar Bamniya<br>Navin Baid |

The funders had no role in study design, data collection and interpretation, or the decision to submit the work for publication.

## Author contributions

Hina Khan, Partha Paul, Harsh Goar, Conceptualization, Data curation, Formal analysis; Bhanwar Bamniya, Navin Baid, Data curation, Formal analysis; Dibyendu Sarkar, Conceptualization, Resources, Formal analysis, Supervision, Investigation, Writing - original draft, Project administration, Writing - review and editing

## Author ORCIDs

Bhanwar Bamniya ⬛ http://orcid.org/0009-0003-6467-678X
Dibyendu Sarkar ⬛ https://orcid.org/0000-0001-6499-177X

## Ethics

All experiments pertaining to mice were in accordance with Institutional regulations after review of protocols and approval by the Institutional Animal Ethics Committee (IAEC/17/05, and IAEC/19/02).

Reviewer #1 (Public review): https://doi.org/10.7554/eLife.92136.4.sa1

Reviewer #2 (Public review): https://doi.org/10.7554/eLife.92136.4.sa2
Author response https://doi.org/10.7554/eLife.92136.4.sa3

## Additional files

### Supplementary files
• Supplementary file 1. Sequences of oligonucleotide primers used in RT-qPCR, ChIP-qPCR, cloning and amplifications. (**a**) Sequences of oligonucleotide primers used in RT-qPCR and ChIP-qPCR measurements reported in this study. (**b**) Sequences of oligonucleotide primers used for amplification and cloning, and plasmids used in this study.
• MDAR checklist

### Data availability
All data generated or analysed during this study are included in the manuscript and supporting files; source data files have been provided for the figures.

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

# Appendix 1

**Appendix 1—key resources table**

| Reagent type (species) or resource | Designation | Source or reference | Identifiers | Additional information |
|---|---|---|---|---|
| Strain, strain background (*Mycobacterium tuberculosis*) | WT-H37Rv | ATCC25618 | | Wild-type *M. tuberculosis* H37Rv strain |
| Strain, strain background (*M. tuberculosis*) | Δ*phoP*-H37Rv | *Walters et al., 2006* | | *phoPR* locus *Rv0757-Rv0758* has been inactivated |
| Strain, strain background (*M. tuberculosis*) | Δ*phoP*::*phoP* | *Walters et al., 2006* | | Δ*phoP*-complemented with *phoP* |
| Sequence-based reagent | FPrv0386RT | This study | RT-qPCR primer | Gene-specific primer: GGCCCAGATCCTTACCTTTC |
| Sequence-based reagent | RPrv0386RT | This study | RT-qPCR primer | Gene-specific primer: TGTGCAGCACTTCCTGAGAC |
| Sequence-based reagent | FPrv0891cRT | This study | RT-qPCR primer | Gene-specific primer: TCAAACGGTACGAGGGTGAT |
| Sequence-based reagent | RPrv0891cRT | This study | RT-qPCR primer | Gene-specific primer: CACAACTGCATGACCCATTC |
| Sequence-based reagent | FPrv1264RT | This study | RT-qPCR primer | Gene-specific primer: CAGCTAGGCGAAGTGGTGTC |
| Sequence-based reagent | RPrv1264RT | This study | RT-qPCR primer | Gene-specific primer: GGGAAAGTTGTTGTCGGTGT |
| Sequence-based reagent | FPrv1339RT | This study | RT-qPCR primer | Gene-specific primer: CGCCGTTGGTTATCGACTTC |
| Sequence-based reagent | RPrv1339RT | This study | RT-qPCR primer | Gene-specific primer: AACAGTCGTCAATCTCCCCA |
| Sequence-based reagent | FPrv1625cRT | This study | RT-qPCR primer | Gene-specific primer: TGAATTTGCCCCACCGAATC |
| Sequence-based reagent | RPrv1625cRT | This study | RT-qPCR primer | Gene-specific primer: CAGCGCAATTGAAGGATCCA |
| Sequence-based reagent | FPrv1647RT | This study | RT-qPCR primer | Gene-specific primer: GCCCAAGATGCTGTGAAGTC |
| Sequence-based reagent | RPrv1647RT | This study | RT-qPCR primer | Gene-specific primer: AACTCACTTTGCGGGATCAG |
| Sequence-based reagent | FPrv2488cRT | This study | RT-qPCR primer | Gene-specific primer: TTGCTGTTGGCATCATGTCT |
| Sequence-based reagent | RPrv2488cRT | This study | RT-qPCR primer | Gene-specific primer: CTTCTGGGCATCATCTAGGC |
| Sequence-based reagent | FPrv0805RT | This study | RT-qPCR primer | Gene-specific primer: GCCGAACTACGCAAATTCTT |
| Sequence-based reagent | RPrv0805RT | This study | RT-qPCR primer | Gene-specific primer: ATCCAAAACACTCGGAATCG |
| Sequence-based reagent | FPrv1357cRT | This study | RT-qPCR primer | Gene-specific primer: TCCTCGTCTACCAGCCAATC |
| Sequence-based reagent | RPrv1357cRT | This study | RT-qPCR primer | Gene-specific primer: GAGACGTTGACGCTGACAAA |

*Appendix 1 Continued on next page*

*Appendix 1 Continued*

| Reagent type (species) or resource | Designation | Source or reference | Identifiers | Additional information |
|---|---|---|---|---|
| Sequence-based reagent | FPrv2837cRT | This study | RT-qPCR primer | Gene-specific primer: AGCAGGACCTTGATGGACAG |
| Sequence-based reagent | RPrv2837cRT | This study | RT-qPCR primer | Gene-specific primer: GTTCGACCTCCTTGAACACC |
| Sequence-based reagent | FPphoPRT | *Khan et al., 2022* | RT-qPCR primer | Gene-specific primer: GCCTCAAGTTCCAGGGCTTT |
| Sequence-based reagent | RPphoPRT | *Khan et al., 2022* | RT-qPCR primer | Gene-specific primer: CCGGGCCCGATCCA |
| Sequence-based reagent | FPpks2RT | *Goyal et al., 2011* | RT-qPCR primer | Gene-specific primer: GTTGTGGAAGGCGTTGTTAC |
| Sequence-based reagent | RPpks2RT | *Goyal et al., 2011* | RT-qPCR primer | Gene-specific primer: GTCGTAGAACTCGTCGCAAT |
| Sequence-based reagent | FPmsl3RT | *Goyal et al., 2011* | RT-qPCR primer | Gene-specific primer: GTGAAAACAAACTTCGGTCAC |
| Sequence-based reagent | RPmsl3RT | *Goyal et al., 2011* | RT-qPCR primer | Gene-specific primer: ACAAAGAGTTCAGTGTCAATCTCAG |
| Sequence-based reagent | FPlipFRT | *Bansal et al., 2017* | RT-qPCR primer | Gene-specific primer: TAGTGGCCATCTCTCCGTTG |
| Sequence-based reagent | RPlipFRT | *Bansal et al., 2017* | RT-qPCR primer | Gene-specific primer: AGCGGCTCATAGAGGTCTTC |
| Sequence-based reagent | FP16SrDNART | *Khan et al., 2022* | RT-qPCR primer | Gene-specific primer: CTGAGATACGGCCCAGACTC |
| Sequence-based reagent | RP16SrDNART | *Khan et al., 2022* | RT-qPCR primer | Gene-specific primer: CGTCGATGGTGAAAGAGGTT |
| Sequence-based reagent | FPrv0805up | This study | ChIP-qPCR primer | Promoter-specific primer: CGGCGTTCTGGTATCTCG |
| Sequence-based reagent | RPrv0805up | This study | ChIP-qPCR primer | Promoter-specific primer: TAAGAGAACGTAATCCGG |
| Sequence-based reagent | FPrv0805start | This study | Cloning primer | Gene-specific primer: AATAATGATATCGTGCATAGACTT |
| Sequence-based reagent | RPrv0805stop | This study | Cloning primer | Gene-specific primer: AATAATAAGCTTTCAGTCGACGGGA |
| Sequence-based reagent | FPrv0805N97A | This study | Cloning primer | Gene-specific primer: TGGGTGATGGGTGCACACGACGACCG |
| Sequence-based reagent | RPrv0805N97A | This study | Cloning primer | Gene-specific primer: CGGTCGTCGTGTGCACCCATCACCCA |
| Sequence-based reagent | FPphoPsg | This study | Guide RNA-specific primer | Gene-specific primer: GGGAGATCCAGCGCCTGTGCCCCG |
| Sequence-based reagent | RPphoPsg | This study | Guide RNA-specific primer | Gene-specific primer: AAACCGGGGCACAGGCGCTGGATC |
| Sequence-based reagent | FPrv0805sg | This study | Guide RNA-specific primer | Gene-specific primer: GGGAGCTCGACCAGGCCTCGGAGC |
| Sequence-based reagent | RPrv0805sg | This study | Guide RNA-specific primer | Gene-specific primer: AAACGCTCCGAGGCCTGGTCGAGC |

*Appendix 1 Continued on next page*

*Appendix 1 Continued*

| Reagent type (species) or resource | Designation | Source or reference | Identifiers | Additional information |
|---|---|---|---|---|
| Sequence-based reagent | FPpRH2521seq | This study | Vector-specific primer | Vector-specific primer: AAACTCTAGAAATATTGGATCG |
| Sequence-based reagent | RPpRH2521seq | This study | Vector-specific primer | Vector-specific primer: CCTAATGACCATGGTGACCTC |
| Recombinant DNA reagent | p19Kpro[b] | *De Smet et al., 1999* | Plasmid DNA | Mycobacteria expression vector, Hyg[r] |
| Recombinant DNA reagent | p19Kpro-*phoP* | *Anil Kumar et al., 2016* | Plasmid DNA | His$_6$-tagged PhoP residues 1–247 cloned in p19Kpro |
| Recombinant DNA reagent | p19Kpro-*phoP*(FLAG) | This study | Plasmid DNA | FLAG-tagged PhoP residues 1–247 cloned in p19Kpro |
| Recombinant DNA reagent | pSTKi[c] | *Parikh et al., 2013* | Plasmid DNA | Integrative mycobacterial expression vector, Kan[rc] |
| Recombinant DNA reagent | pSTKi-*pde* | This study | Plasmid DNA | PDE (*rv0805*) residues 1–957 cloned in pSTKi |
| Recombinant DNA reagent | pSTKi-*pdeM* | This study | Plasmid DNA | PDE (*rv0805M*) Asn-97 codon mutated to Ala in pSTKi-*pdeM* |
| Recombinant DNA reagent | pRH2502 | *Singh et al., 2016* | Plasmid DNA | Integrative mycobacterial expression vector, Kan[r, c] |
| Recombinant DNA reagent | pRH2521 | *Singh et al., 2016* | Plasmid DNA | Episomal expression vector, Hyg[r,b] |
| Recombinant DNA reagent | pRH2521-phoPsg | This study | Plasmid DNA | pRH2521 vector expressing *phoP* guide RNA, Hyg[r,b] |
| Recombinant DNA reagent | pRH2521-rv0805sg | This study | Plasmid DNA | pRH2521 vector expressing *rv0805* guide RNA, Hyg[r,b] |
| Antibody | Anti-GroEL2 (rabbit polyclonal) | Abcam | Cat# ab90522 | 1:5000 |
| Antibody | Anti-RpoB (rabbit monoclonal) | Abcam | Cat# ab191598 | 1:3000 |
| Antibody | Anti-CFP10 (rabbit polyclonal) | Abcam | Cat# ab45074 | 1:1000 |

