## [Editor Report · eLife assessment]

This **important** study describes how PhoP regulates cyclic-AMP production in the human pathogen *Mycobacterium tuberculosis*. The authors provide **convincing** evidence that PhoP acts as a repressor of the cyclic-AMP-specific phosphodiesterase, Rv0805, which can degrade cyclic-AMP. The article has addressed all outstanding comments, and the work will be of interest to bacteriologists.

---

## [Referee Report · Reviewer #1 (Public review)]

Summary:

This paper provides a straightforward mechanism of how mycobacterial cAMP level is increased under stressful conditions and shows that the increase is important for the survival of the bacterium in animal hosts. The cAMP level is increased by decreasing the expression of an enzyme that degrades cAMP.

Strengths:

The paper shows that under different stresses the response regulator PhoP represses a phosphodiesterase (PDE) that degrades cAMP specifically. Identification of PhoP as a regulator of cAMP is significant progress in understanding Mtb pathogenesis, as an increase in cAMP apparently increases bacterial survival upon infection. On the practical side, reduction of cAMP by increasing PDE can be a means to attenuate the growth of the bacilli. The results have wider implications since PhoP is implicated in controlling diverse mycobacterial stress responses and many bacterial pathogens modulate host cell cAMP levels. The results here are straightforward, internally consistent, and of both theoretical and applied interests. The results also open considerable future work, especially how increases in cAMP level help to increase survival of the pathogen.

Weaknesses:

It is not clear whether PhoP-PDE Rv0805 is the only pathway to regulate cAMP level under stress.

Comments on revised submission:

The authors have addressed my comments adequately, actually except for all but one. I have only one comment to do with the last line of the abstract. First, "genetic manipulation" usually means changing DNA. In Mtb pathogenesis I hope there is no DNA modification or change in the bacterial DNA. Also, the authors did not really inactivate the whole PhoP- rv0805-cAMP pathway. It would be best if the last line is made more fact based: Thus, inactivation of PhoP decreases cAMP level, thereby stress tolerance and intracellular survival of the bacillus.

---

## [Referee Report · Reviewer #2 (Public review)]

Summary:

In the manuscript, the authors have presented new mechanistic details to show how intracellular cAMP levels are maintained and linked to the phosphodiesterase enzyme which in turn is controlled by PhoP. Later, they showed the physiological relevance linked to altered cAMP concentrations.

Strengths:

Well-thought-out experiments. The authors carefully planned the experiments well to uncover the molecular aspects of it diligently.

Weaknesses:

None. The authors have meticulously responded to all my queries and concerns through multiple rounds of review.

---

## [Author Response]

The following is the authors’ response to the previous reviews.

**Reviewer #1 (Public Review):**
Summary:This paper provides a straightforward mechanism of how mycobacterial cAMP level is increased under stressful conditions and shows that the increase is important for the survival of the bacterium in animal hosts. The cAMP level is increased by decreasing the expression of an enzyme that degrades cAMP.

We thank the reviewer for these extremely encouraging comments.

Strengths:The paper shows that under different stresses the response regulator PhoP represses a phosphodiesterase (PDE) that degrades cAMP specifically. Identification of PhoP as a regulator of cAMP is significant progress in understanding Mtb pathogenesis, as increase in cAMP apparently increases bacterial survival upon infection. On the practical side, reduction of cAMP by increasing PDE can be a means to attenuate the growth of the bacilli. The results have wider implications since PhoP is implicated in controlling diverse mycobacterial stress responses and many bacterial pathogens modulate host cell cAMP level. The results here are straightforward, internally consistent, and of both theoretical and applied interests. The results also open considerable future work, especially how increases in cAMP level help to increase survival of the pathogen.Weaknesses:It is not clear whether PhoP-PDE Rv0805 is the only pathway to regulate cAMP level under stress.
**Reviewer 1 (Recommendations for the authors):**
(1) L.1: "maintenance of" or 'regulating'- I thought change in cAMP level upon stress is the whole point of the paper. Also, can replace "intracellular survival" with 'survival in host macrophages' if you want to be more specific.

We agree with the reviewer, and therefore, we have now replaced “maintenance of” with “regulating cAMP level” in the title. However, we feel more comfortable with “intracellular survival” rather than being more specific with ‘survival in host macrophages’ as we have also shown animal experiments to demonstrate ‘in vivo’ effect in mice lung and spleen.

(2) L.26: ---requires the bacterial virulence regulator –

The suggested change has been made to the text.

(3) L.30: Replace "phoP locus since the" with 'PhoP since this'. (The product, not the locus, is the regulator). The same comment for l.113.

We agree with the reviewer. The suggested changes have been made to the text.

(4) L.31: Change represtsor to repressor.

We are sorry for the embarrassing spelling mistake. We have rectified the mistake in the revised version.

(5) L.32: "hydrolytically degrades" or hydrolyses? (lytic and degrade sound like tautology). Same comment for l.117.

We agree. The suggested change has been made to the text in both places of the revised manuscript.

(6) L.35: I would also suggest changing "intra-mycobacterial" to 'intra bacterial' because you are talking about one bacterium here. The same change is recommended in l.29.

Following reviewer’s recommendation, we have made the changes in the revised manuscript.

(7) L.37: bacillus unless use of the plural form is the norm in the field.

We agree. The suggested change has been made to the text.

(8) L.43: Delete "intracellular" and change "intracellular" to host in l.44.

The suggested changes have been made to the text.

(9) L.66: --that a burst--

We have corrected the mistake in the revised manuscript.

(10) L.76: Receptor or receptor?

We have corrected the mistake in the revised manuscript.

(11) L.86: -- mechanisms of regulation of mycobacterial cAMP level. (homeostasis needs to be introduced first, and not used in the concluding statement for the first time).

The suggested changes have been made to the text.

(12) L.96: "essential" or 'a requirement'. (reduction is not the same as elimination)

We understand the reviewer’s concern. However, several studies have independently established that phoPR remains an essential requirement for mycobacterial virulence.

(13) L.97: Moreover, a mutant

The suggested change has been made to the text.

(14) L.113: --locus since PhoP has been –

The suggested change has been made to the text.

(15) L.119: mechanism or manner? (you are stating a fact, not a mechanism)

We agree. We have now replaced ‘mechanism’ with ‘manner’ in the revised manuscript.

(16) L.130: --lacking copies of both phoP and phoR (I am assuming you don't have two copies of each gene)

We understand the reviewer’s concern. For better clarity, we have now clearly mentioned that the phoPR-KO mutant lacks both the single copies of phoP and phoR genes.

(17) L.156: Indicate why GroEL2? - cells as another cytoplasmic protein, GroEL2 was also undetectable

We have now mentioned it in the secretion experiments that mycobacterial cells did not undergo autolysis. To prove this point, we have used cytoplasmic GroEL2 as a marker protein. The absence of detectable GroEL2 in the culture filtrates (CFs) suggests absence of autolysis. To this end, we have modified the sentence in the revised manuscript (duplicated below):

“Fig. 1C confirms absence of autolysis of mycobacterial cells as GroEL2, a cytoplasmic protein, was undetectable in the culture filtrates (CF).”

(18) L.266: May delete "Together". Start with These data--, which would draw more attention to integrated view. In l.268-270, a reminder that intracellular pH is acidic in the normal course would enhance the physiological significance of the present results.

We agree. We have made the suggested changes to the text. In view of the second comment of the reviewer, we have modified the text (duplicated below):

“These data represent an integrated view of our results suggesting that PhoP-dependant repression of rv0805 regulates intra-mycobacterial cAMP level. In keeping with these results, activated PhoP under acidic pH conditions significantly represses rv0805, and intracellular mycobacteria most likely utilizes a higher level of cAMP to effectively mitigate stress for survival under hostile environment including acidic pH of the phagosome.”

(19) L.272: Delete "and intracellular survival" (?) (I am assuming the survival is due to stress tolerance; also the section talks about stress only). No period in l.273.

Following reviewer’s recommendations, the suggested changes have been made to the text.

(20) L.295: Start the sentence thus: It appears that at least one of ---. (This would put more emphasis on the inference)

We agree. We have now incorporated the recommended changes in the revised version.

(21) L.301: No parenthesis.

The parenthesis has been removed in the revised manuscript.

(22) L.306: Together already implies these. Either delete Together (which I would prefer) or say 'Together, the results suggest that strains expressing wild type and mutant----properties, and the results are

We agree. We have now deleted ‘Together’ in the revised manuscript.

(23) L.311: These results support our view that higher---- (to avoid repetition of l.266)

We agree. We have now incorporated the suggested change in the revised manuscript.

(24) L.316: Using or with?

We think “with” goes well with the statement.

(25) L.329: Rephrase thus: Effect of intra-bacterial cAMP level on in vivo--

The recommended change has been made to the text.

(26) L.333: I would use ~, if you want to indicate about.

We agree. We have now used ‘~’ in the revised version. Changes were incorporated in lines 328, 330 and 333 of the revised manuscript.

(27) L.350: Change "somewhat functionally" to phenotypically?

We thank the reviewer for this suggestion. We have changed “somewhat functionally” to “phenotypically” in the revised manuscript.

(28) L.361: Change "is connected to" to 'regulates'.

The suggested change has been made to the text.

(29) L.365: ACs (to be parallel with PDEs)

We agree. The suggested change has been made to the text.

(30) L.366: delete "very" (let the readers decide how recent from the reference date).

The suggested change has been made to the text.

(31) L.382: level remained unknown before the present study.

The recommended change has been made to the text.

(32) L.399: add at the end of the sentence 'under stress'. Also, represent, not represents.

The recommended changes have been made to the text.

(33) L.560 and 571: Section headings formatted differently from the rest. Similar problem in l.900.

We have rectified the issue and all of the section headings are now formatted in the same style.

**Reviewer #2 (Public Review):**
Summary:In the manuscript, the authors have presented new mechanistic details to show how intracellular cAMP levels are maintained linked to the phosphodiesterase enzyme which in turn is controlled by PhoP. Later, they showed the physiological relevance linked to altered cAMP concentrations.Strengths:Well thought out experiments. The authors carefully planned the experiments well to uncover the molecular aspects of it diligently.

We thank the reviewer for these extremely encouraging comments.

Weaknesses:Some fresh queries were made based on the author's previous responses and hope to get satisfactory answers this time.

We provide below a point-by-point response to the fresh queries.

(2) Line 134: please describe the complementation strain features as it is mentioned for the first time (plasmid, copy number, promoter etc.) in the manuscript. Especially under NO stress what could be the authors' justification regarding the high cAMP concentration in the complementation strain?As recommended by the reviewer, the details of construction of the complemented strain have been incorporated in the 'Materials and Methods' section of the revised manuscript (duplicated below): "To complement phoPR expression, pSM607 containing a 3.6-kb DNA fragment of M. tuberculosis phoPR including 200-bp phoP promoter region, a hygromycin resistance cassette, attP site and the gene encoding phage L5 integrase, as detailed earlier (Walters et al., 2006) was used to transform phoPR mutant to integrate at the L5 attB site." To address the reviewer's other concern, we have now included the following sentence in the 'Results' section of the revised manuscript (duplicated below): "A higher cAMP level in the complemented strain under NO stress is possibly attributable to reproducibly higher phoP expression in the complemented mutant under specific stress condition (Khan et al., 2022)."Reference: Khan et al. (2022) Convergence of two global regulators to coordinate expression of essential virulence determinants of Mycobacterium tuberculosis. eLife 2022, 11:e80965.New query: The complemented gene (in pSM607 plasmid) becomes a single copy after chromosomal integration, so it should ideally behave like a WT strain. How could authors still justify the high cAMP concentration under NO stress?

We agree with the reviewer. We are unable to provide a cogent justification regarding this result. We speculate that PhoP is strikingly activated under NO stress by a non-canonical mechanism and strongly represses rv0805 expression. As a result, there is a significantly higher cAMP concentration in case of the complemented mutant under NO stress.

(13) Line 292: There is a difference between red and green bars. Authors should do statistical analysis and then comment on whether overexpression of WT and mutant pde are different or similar, to me they are different; also, explain why the WT-Rv0805 strain is different than the phoPR-KO strain in the context of cell wall metabolism.As recommended by the reviewer, we have now included statistical significance of the data in the revised version, and modified the text accordingly in the manuscript.New query: Authors are asked to put a statistical significance test between WT-Rv0805 and WT-Rv0805M.

We have included it in the modified figure. Also, to explain it we incorporated new text in the legend to Fig. 4C of the revised manuscript (duplicated below):

“Note that similar to phoPR-KO, WT-Rv0805 shows a comparably higher sensitivity to CHP relative to WT bacilli. However, WT-Rv0805M expressing a mutant Rv0805, shows a significantly lower sensitivity to CHP relative to WT-Rv0805, as measured by the corresponding CFU values.”

(14) Line 299-303: Authors should explain how the colocalization % are calculated. Also, in the figure 4D merge panel please highlight the difference.As suggested by the reviewer, we have now explained the methodology used to calculate percent colocalization in greater details. Also, we have modified Figure 4D to highlight the difference between samples shown in merge panel. Please see our response to comment # 33 from the Reviewer 1.New query: In the figure legend it should be mentioned that the white arrow indicates non-co-localization which is visibly higher in WT and WT Rvo805M.

We thank the reviewer for this very important suggestion. We have now included the following text in the legend to Fig. 4D of the revised manuscript.

“White arrowheads in the merge panels indicate non-colocalization, which remains higher in WT-H37Rv and WT-Rv0805M relative to phoPR-KO or WT-Rv0805.”